# Senescence in yeast is associated with amplified linear fragments of chromosome XII rather than ribosomal DNA circle accumulation

**Andre Zylstra**[¤a], **Hanane Hadj-Moussa**[ʘ], **Dorottya Horkai**[¤bʘ], **Alex J. Whale**, **Baptiste Piguet**, **Jonathan Houseley***

Epigenetics Programme, Babraham Institute, Cambridge, United Kingdom

ʘ These authors contributed equally to this work.
¤a Current address: Molecular Systems Biology, Groningen Biomolecular Sciences and Biotechnology Institute, University of Groningen, Groningen, the Netherlands
¤b Current address: Abcam plc, Biomedical Campus, Cambridge, United Kingdom
* jon.houseley@babraham.ac.uk

**Data Availability Statement:** All RNA-seq data has been deposited at GEO under accession number GSE207429. All other relevant data are within the

## Abstract

The massive accumulation of extrachromosomal ribosomal DNA circles (ERCs) in yeast mother cells has been long cited as the primary driver of replicative ageing. ERCs arise through ribosomal DNA (rDNA) recombination, and a wealth of genetic data connects rDNA instability events giving rise to ERCs with shortened life span and other ageing pathologies. However, we understand little about the molecular effects of ERC accumulation. Here, we studied ageing in the presence and absence of ERCs, and unexpectedly found no evidence of gene expression differences that might indicate stress responses or metabolic feedback caused by ERCs. Neither did we observe any global change in the widespread disruption of gene expression that accompanies yeast ageing, altogether suggesting that ERCs are largely inert. Much of the differential gene expression that accompanies ageing in yeast was actually associated with markers of the senescence entry point (SEP), showing that senescence, rather than age, underlies these changes. Cells passed the SEP irrespective of ERCs, but we found the SEP to be associated with copy number amplification of a region of chromosome XII between the rDNA and the telomere (ChrXIIr) forming linear fragments up to approximately 1.8 Mb size, which arise in aged cells due to rDNA instability but through a different mechanism to ERCs. Therefore, although rDNA copy number increases dramatically with age due to ERC accumulation, our findings implicate ChrXIIr, rather than ERCs, as the primary driver of senescence during budding yeast ageing.

## Introduction

Replicative ageing in the budding yeast *Saccharomyces cerevisiae* is a widely used model of ageing in dividing cells [1–3]. Under this paradigm, age is the number of budding events undergone by a mother cell, and life span is the total number of divisions prior to a permanent loss

paper and its Supporting information files. Detailed and updated protocols are available at https://www.babraham.ac.uk/our-research/epigenetics/jon-houseley/protocols.

**Funding:** JH and AZ were funded by the Wellcome Trust [110216], AZ and DH by BBSRC DTP PhD awards [1645489, 1947502], JH, AW and HHM by the BBSRC [BI Epigenetics ISP: BBS/E/B/000C0423], BP by Ecole Normale Superieure Paris-Saclay, Universite Paris-Saclay. The funders had no role in study design, data collection and analysis, decision to publish, or preparation of the manuscript.

**Competing interests:** The authors have declared that no competing interests exist.

**Abbreviations:** BH, Benjamini–Hochberg; ERC, extrachromosomal ribosomal DNA circle; ESR, environmental stress response; FDR, false discovery rate; MEP, mother enrichment program; PCA, principal component analysis; rDNA, ribosomal DNA; RFB, replication fork barrier; SEP, senescence entry point; WGA, wheat germ agglutinin.

of replicative capacity [4]. Long-term imaging shows that cell cycle duration remains constant for most of the replicative lifetime but lengthens dramatically in the last few divisions [5–7]. This indicates a period of pathological ageing that precedes loss of replicative viability, although the cause of this transition to a pathological ageing state is still debated.

Ageing cells display a spectrum of molecular phenotypes including dramatic changes in gene expression. This has been attributed to a loss of transcriptional repression due to histone depletion [8,9], the induction of a widespread but defined environmental stress response (ESR) [10–13] and the uncoupling of translation from mRNA abundance [14]. Whatever the cause, ageing yeast undergo widespread induction of coding and noncoding loci that are normally repressed for RNA polymerase II transcription during vegetative growth and increase expression of stress and environmental response factors while the abundance of mRNAs encoding translation and ribosome biogenesis components is decreased [8,11,12,15,16]. Similarly, and perhaps consequently, the proteome becomes dysregulated, aggregation prone, and increasingly uncoupled from the transcriptome; gene expression becomes unresponsive; and the metabolome shifts towards reduced growth and substrate uptake [14,17–19]. Interestingly, single-cell techniques have revealed that the onset of senescence is a discrete process initiated abruptly at the senescence entry point (SEP) [5,20]. After the SEP, cells exhibit an extended G1 phase marked by intense foci of the tagged mitochondrial membrane protein Tom70-GFP and low levels of cyclin Cln1. However, cell growth does not pause, leading to rapid cell size expansion and pathological cytoplasmic dilution coincident with nuclear and nucleolar growth [5,6,17,20,21]. One or more of these phenotypes may cause the permanent loss of replicative potential that defines yeast replicative life span, and the reproducible time from the SEP to loss of replicative viability suggests that senescence limits life span to some extent [5], but we and others have observed ageing trajectories resulting in loss of replicative potential without apparent senescence [22,23], so other non-SEP mechanisms must also limit life span and the causal connection between the SEP and loss of viability remains unproven.

The most prominent theory of yeast ageing focuses on accumulation of ERCs, which arise during homologous recombination within the highly repetitive ribosomal DNA (rDNA) locus [24–26]. ERCs are replicated in each cell cycle and are asymmetrically retained in mother cells at mitosis by the SAGA and TREX-2 complexes, such that ERC copy number rises exponentially during ageing to the point that effective genome size increases 30% to 40% in 24 hours [15,20,24,27,28]. Due to this behaviour, ERCs are very plausible "ageing factors" that arise early in ageing and later cause nucleolar fragmentation [24,29,30]. Important components of this theory are Sir2, the eponymous member of the highly conserved sirtuin protein deacetylase family, and Fob1, which localises to the rDNA replication fork barrier (RFB) site. Fob1 stalls replication forks at the RFB and averts head-on collision with RNA polymerase I, but resolution of the stalled forks can be recombinogenic through repair events leading to ERC formation [26,31,32]. In support of the ERC theory, *SIR2* deletion destabilises the rDNA, promotes ERC accumulation and rDNA copy number variation, and shortens yeast life span, whereas *FOB1* deletion or *SIR2* overexpression extend life span [26,33]. ERC-encoded noncoding RNAs are abundantly expressed by RNA polymerase II with a concomitant accumulation of activating epigenetic marks such as H3K4me3, while the additional ribosomal DNA genes are transcribed by RNA polymerase I [15,16,20]. As such, it is not hard to imagine that titration of nuclear and nucleolar factors by thousands of ERCs could imbalance the protein coding transcriptome and/or ribosome synthesis.

Anti-ageing interventions with cross-species efficacy, such as caloric restriction and mTOR inhibition, also reduce rDNA instability and recombination through Sir2 [34–37]. Remarkably, there are indications that rDNA instability and copy number variation might be relevant to animal ageing and pathology as rDNA copy number variation has been reported during

*Drosophila* and mouse ageing [38,39], in human neurodegenerative disorders [40], and in cancer [41–44]. Circular DNAs containing rDNA sequence have also been observed in *Drosophila* and human cells [45,46]. However, although accumulation of ERCs in ageing yeast was first reported a quarter of a century ago, there remains a conspicuous lack of analogous data for any other model organism [24]. Nonetheless, recent publications showing a tight connection between ERC accumulation and ageing phenotypes in yeast strongly suggest that ERCs are the proximal cause of age-related cell cycle disruption and senescence. Neurohr and colleagues observed that post-SEP mother cells are rejuvenated through rare events in which the entire ERC complement is transmitted to a daughter cell; Ünal and colleagues observed that ERCs are compartmentalised and degraded during meiosis in aged cells [47,48]; while Morlot and colleagues reported that cell cycle disruption and the SEP coincide with exponential ERC accumulation and concomitant excess rDNA transcription [19,20]. Furthermore, Meinema and colleagues recently reported that extended association of ERCs with nuclear pore complexes can promote the progressive decay of nuclear pores that occurs during ageing in yeast [49,50].

These studies strongly suggest that ERC accumulation is pathological, but there is little mechanistic evidence of how ERC-dependent pathology may be mediated. To study this, we used mutants in ERC formation and retention to define the effect of ERCs on mRNA abundance and the SEP but unexpectedly found no connection with ERCs. Instead, we show a strong correlation between amplification of a large linear fragment of chromosome XII (ChrXIIr) and the SEP, suggesting that this may be the unanticipated cause of senescence in yeast.

## Results

To investigate the role of ERC accumulation during replicative ageing in *S. cerevisiae*, we examined aged mutant cells that fail to accumulate ERCs during ageing. Homologous recombination-deficient *rad52Δ* cells cannot form ERCs, whereas *spt3Δ* and *sac3Δ* cells lack components of SAGA and TREX-2 so cannot retain ERCs in mother cells (Fig 1A) [25,28]. By studying mutants in different pathways required for age-linked ERC accumulation, we aimed to separate effects caused by ERCs from unrelated effects of individual mutations.

For purification of aged cells, we employed the mother enrichment program (MEP) devised by Lindstrom and Gottschling [51]. MEP cells proliferate normally until β-estradiol is added to a culture, after which all newborn daughter cells are rendered inviable. This allows mother cells to reach advanced replicative age in batch culture without proliferation of young cells depleting media components. Mother cell walls are labelled with reactive biotin when young, which allows recovery of the same cells on magnetic streptavidin beads at aged time points after fixation (Fig 1B). Unlike most ageing systems in yeast, the MEP uses diploid cells, which are more resilient both to random mutations that could circumvent the MEP and to toxic double-strand breaks that occur increasingly with age and result (in diploids) in loss of heterozygosity [51,52].

Not all mutations that alter ERC abundance in young cells have equivalent effects in aged cells. For example, young *fob1Δ* mutant cells contain fewer ERCs, but ERC abundance in aged diploid *fob1Δ* MEP cells hardly differs from wild type (S1A Fig) [52]. We therefore confirmed ERC abundance in all 3 mutants by Southern blot. Reassuringly, deletion of *RAD52* in the MEP background abrogated ERC accumulation (30× reduction) [25,26], while deletion of *SPT3* or *SAC3* dramatically reduced ERC accumulation during ageing relative to wild type (10× and 5× reduction, respectively) (S1B Fig) [28].

Together, *rad52Δ*, *spt3Δ*, and *sac3Δ* form a set of 3 mutants that accumulate very few ERCs during ageing compared to wild type. Consistent differences that emerge during ageing

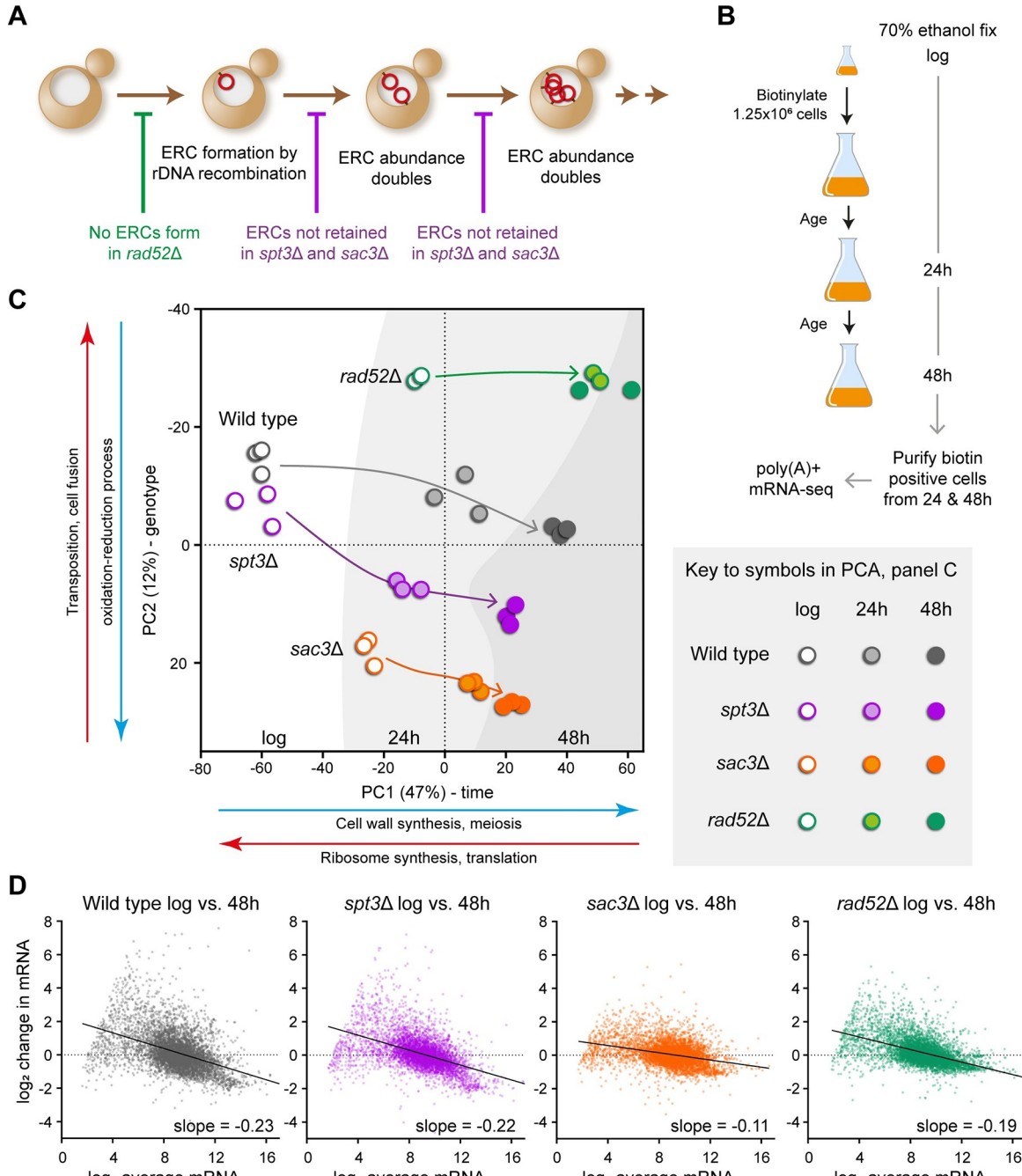

**Fig 1. ERCs do not cause genome-wide gene expression dysregulation.** (**A**) Schematic of ERC accumulation during replicative ageing, indicating the processes that are impaired in *rad52Δ*, *spt3Δ*, and *sac3Δ* mutants. (**B**) Schematic of the MEP culture system used to isolate aged populations for analysis. Times of biotin labelling and harvest are shown. (**C**) PCA summary for young and aged poly(A)+ RNA-seq libraries of wild-type and indicated ERC mutants aged in YPD using the MEP. Arrows show progression across time in culture; background colour graduations indicate sample time point. Major GO categories for each PC are indicated based on the 300 highest rotation genes underlying the PC in either direction, full GO term enrichment analysis in S7 File. (**D**) MA plots comparing $log_2$ mRNA abundance distributions between log phase and 48-hour-aged samples from wild-type and indicated mutants. x-Axis is $log_2$ average normalised read count; y-axis is change in $log_2$ normalised read count from young to old. Slope is calculated by linear regression. Data for each genotype and age calculated as the $log_2$ mean normalised read counts per gene of 3 biological replicates for wild-type, *spt3Δ*, and *sac3Δ* or 2 biological replicates for *rad52Δ*. The numerical data underlying this Figure can be found in S8 File. ERC, extrachromosomal ribosomal DNA circle; GO, gene ontology; MEP, mother enrichment programme; PC, principal component; PCA, principal component analysis.

between each mutant and the wild-type form candidate effects that could be attributable to ERCs, whereas differences observed in only a single mutant or that are already present in young cells prior to ERC accumulation must be attributed to genotype effects rather than ERCs. We note that *rad52Δ* and *sac3Δ* are short lived, whereas the life span of *spt3Δ* is almost identical to wild type, providing a spectrum of lifespans to allow separation of terminal phenotypes from effects of ERCs [25,53,54].

## Age-linked transcriptomic dysregulation is independent of ERCs

The RNA polymerase II transcriptome should provide a sensitive readout of the impact of ERCs on the ageing process, reporting both changes in overall distribution of mRNA abundance and also the expression of individual genes. Individual gene expression differences were predicted to be particularly informative, indicating stress response pathways and/or metabolic feedback circuits activated due to pathological impacts of ERC accumulation.

Poly(A)+ RNA-seq libraries were derived from MEP wild-type, *spt3Δ*, and *sac3Δ* cells in triplicate and *rad52Δ* cells in duplicate. Cells were harvested at log phase and after 24 and 48 hours of ageing, times at which approximately 100% and approximately 30% to 40% of wild-type cells remain viable, respectively (Fig 1B) [15,51]. Expression of 1,186 genes enriched for carbohydrate metabolism, respiration, and cell wall organisation increased significantly at 48 hours compared to log phase in wild type, while 870 genes enriched for ribosome synthesis and translation decreased significantly, all of which is highly consistent with previous ageing gene expression studies and supports the reliability of our dataset (DESeq2, Benjamini–Hochberg (BH) corrected false discovery rate (FDR) < 0.05, $\log_2$-fold change threshold ± 0.5) (S1C Fig) [8,12,15].

Our strategy of defining the effects of ERCs during ageing based on similar gene expression changes in otherwise disparate mutants that all lack ERCs (*spt3Δ*, *sac3Δ*, and *rad52Δ*) relies on these mutations having different effects on gene expression in young cells. In support of this, principal component analysis (PCA) performed on the log phase samples shows no association between any pair of mutants (S1D Fig). In more detail, pairwise differential expression analysis at log phase uncovered >1,000 significantly differentially expressed genes in each mutant compared to the wild type (DESeq2, BH-corrected FDR < 0.01, $\log_2$-fold change threshold ± 0.5), and of the 2,606 genes that are significantly differentially expressed in at least 1 mutant, only 197 (7.5%) change in the same direction in all 3 mutants (93 up, 104 down). Hierarchical clustering of the 2,606 genes also shows little overlap between the effects of the different mutants (S1E Fig). Notably, *spt3Δ* and *sac3Δ*, which might be expected to have similar transcriptomic effects given that Spt3 and Sac3 are physically associated and link promoters to the nuclear pore, actually have almost diametrically opposite effects on gene expression.

We then performed PCA on the full RNA-seq dataset and found that samples segregated by time on PC1 (47% of variance) and by genotype on PC2 (12% of variation). PC1 score increased with age in all genotypes, and the progression was strikingly similar between wild type and *spt3Δ* (Fig 1C, compare grey and purple). The 24- and 48-hour transcriptomes of *rad52Δ* cells clustered together, consistent with these very short-lived cells having reached the end of their replicative life span by 24 hours [25], though the yield of cells did not decrease from 24 to 48 hours and the increased chronological age of the nondividing *rad52Δ* cells evidently did not impact the transcriptome (Fig 1C, green). In contrast, the *sac3Δ* transcriptome progressed on PC1 both from log to 24 and from 24 to 48 hours, but less so than wild type, consistent with slower but ongoing division (Fig 1C, orange).

Yeast cells are reported to undergo genome-wide transcriptional derepression during ageing, particularly of low-expressed genes [8]. This is clearly visible in MA plots, which display

the change in expression with age of each gene as a function of the average expression of that gene (Fig 1D). In wild type, genes with low average expression are strongly induced with age, genes with intermediate expression change little relative to average, and abundant mRNAs decrease relative to average, imparting a characteristic skew away from horizontal to the distribution (Fig 1D, dotted line). The MA plot for *spt3Δ* is essentially identical to wild type (Fig 1D, compare grey and purple), and a similar effect is observed in *rad52Δ* and *sac3Δ* (Fig 1D, compare grey to green and orange).

We therefore conclude that the dramatic remodelling of mRNA abundance which accompanies yeast replicative ageing is completely independent of the massive genomic and epigenomic changes resulting from ERC accumulation.

## ERC accumulation exerts minimal effects on individual genes

Bulk transcriptome properties could mask differences in the behaviour of large numbers of genes. However, for the set of genes significantly differentially expressed between log phase and 48 hours in wild type (S1C Fig), the direction and magnitude of mRNA abundance change during ageing correlates strongly between wild type and mutants (Fig 2A). This is highly consistent with other studies showing common age-related transcriptional patterns between wild type and various mutant genotypes [12,15]. This analysis further shows that broad age-related gene expression changes occur regardless of ERC accumulation.

We next looked for changes in more specific processes or pathways that might be ERC dependent. We performed standard pairwise differential expression analyses between wild type and each mutant in 48-hour-aged samples and looked for genes consistently over- or underexpressed in all mutants lacking ERCs (DESeq2, BH-corrected FDR < 0.05, $\log_2$-fold change threshold ± 0.5). However, this very conservative approach only identified 10 genes, and, worryingly, one of these was *TRP1*, the marker used for deleting *SPT3*, *SAC3*, and *RAD52*. We therefore turned to a more powerful linear modelling approach, and to control for *TRP1* auxotrophy added 3 replicate ageing time-courses for a MEP wild-type strain with the *TRP1* locus repaired.

A linear modelling approach to differential expression analysis [55] considers many datasets together and attempts to explain the differences in expression of each gene between the datasets based on multiple independent variables. For example, in a dataset where age and genotype vary between samples, linear modelling would estimate separate contributions (model coefficients) of age and genotype to the change in expression of each gene; some genes may be more affected by age and have a higher age coefficient, others by genotype and have a higher genotype coefficient. The significance of the contribution of each variable to the differential expression of each gene is also calculated, allowing the set of genes significantly affected by each experimental variable to be extracted. This means that linear modelling can identify genes particularly influenced by one (or more) variable(s) in complex experiments that go beyond simple pairwise comparisons.

We used 4 categorical independent variables in our model: Genotype (wild type, *spt3Δ*, *rad52Δ*, *sac3Δ*), TRP1 (present, not present), Age (log phase, 24 hours, 48 hours), ERC abundance (high, low). Aged wild-type samples were classified as "high" ERC abundance, and all other samples were classified as "low" since aged *spt3Δ*, *rad52Δ*, and *sac3Δ* cells show little accumulation of ERCs above log phase (S1B Fig). Of 161 genes significantly differentially expressed between "high" and "low" ERC abundance (DESeq2, BH-corrected FDR < 0.05, $\log_2$-fold change threshold ± 0.5), only 3 genes (*CSS1*, *PHO11*, and *PHO12*) were significantly down-regulated, while 158 genes were significantly up-regulated in high ERC abundance samples and therefore classified as "ERC-induced." Six genes were differentially expressed

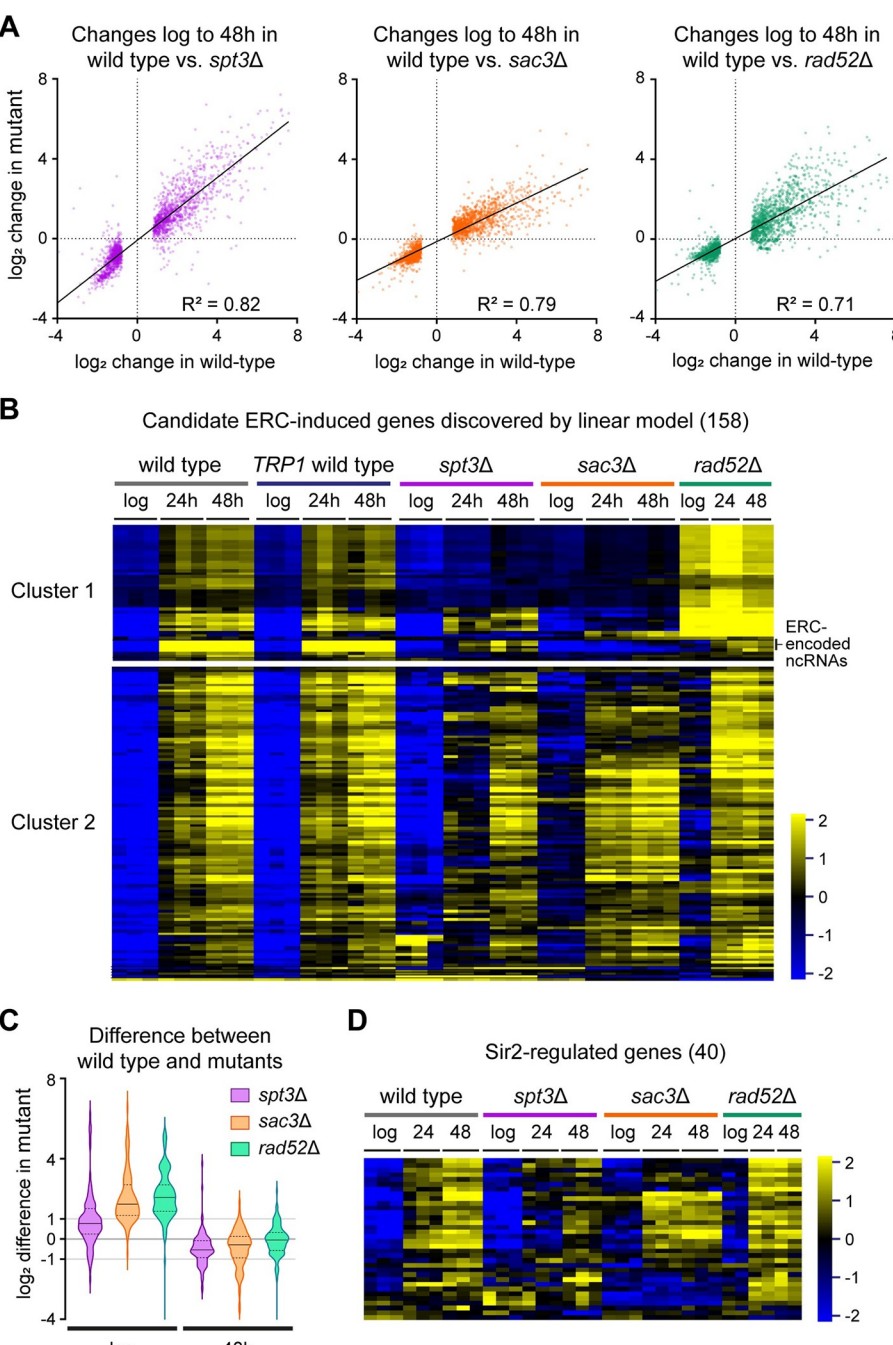

**Fig 2. Expression change of individual genes is not associated with ERC accumulation.** (**A**) Plots of change in log$_2$ mRNA abundance between log phase and 48-hour-aged cells for each individual mutant compared to wild type. Only the 1,186 genes called as significantly differentially expressed with age in wild type (DESeq2 pairwise comparison of wild-type 48-hour-aged vs. log phase, BH-corrected FDR < 0.05, log$_2$-fold change threshold ±0.5) are included. Slopes calculated by linear regression. Data for each genotype and age calculated as the log$_2$ mean normalised read counts per gene of 3 biological replicates for wild type, *spt3Δ*, and *sac3Δ* or 2 biological replicates for *rad52Δ*. Log$_2$ change calculated as the difference between log$_2$ means for 48 hour and log phase samples within a genotype. (**B**) Hierarchical clustering of log$_2$ normalised mRNA abundance for the 158 genes significantly differentially up-regulated in the presence of ERCs based on Linear Model 1 (DESeq2 linear model factor "high" vs. "low" ERC abundance BH corrected (FDR) < 0.05, log$_2$-fold change threshold ±0.5). The colour scale indicates the log$_2$ fold change relative to the per-gene median. Individual biological replicates shown (6 for wild type; 3 for *spt3Δ* and *sac3Δ*; 2 for *rad52Δ*). Wild-type cells with *TRP1* gene repaired are included in addition to standard MEP wild type and are almost identical. Full output of the model is provided in S1 File. (**C**) Violin plot for the difference in log$_2$ mean normalised mRNA

abundance between wild type and given mutants at log phase and after 48 hours of ageing for the cluster 2 genes shown in (**B**). Solid horizontal lines within violins indicate median values, while lower and upper quartiles are indicated by dotted horizontal lines. (**D**) Hierarchical clustering of $\log_2$ normalised mRNA abundances for the set of 40 Sir2-regulated genes defined by Ellahi and colleagues [62]. Individual biological replicates shown (3 for wild type, *spt3Δ*, and *sac3Δ*, and 2 for *rad52Δ*). Clustering procedure and scale as in (**B**). The numerical data underlying this Figure can be found in S8 File. BH, Benjamini–Hochberg; ERC, extrachromosomal ribosomal DNA circle; FDR, false discovery rate; MEP, mother enrichment program.

depending on *TRP1*, but these did not overlap with genes affected by ERC abundance (Linear Model 1, S1 File).

Following hierarchical clustering, we split the 158 candidate ERC-induced genes into 2 large clusters, which most noticeably differ in the *rad52Δ* dataset (Fig 2B). Cluster 1 genes were mostly highly expressed in *rad52Δ* and low in *spt3Δ* and *sac3Δ* (Fig 2B, top). This cluster includes annotations covering the regions of chromosomal rDNA and ERCs that are normally silenced by Sir2 [56–59]. Age-related induction of RNA polymerase II–transcribed noncoding RNAs within rDNA repeats was very low in all 3 low-ERC mutants but increased dramatically with age in wild type, coherent with high transcriptional activity of these regions on ERCs (Figs 2B and S2A) [16]. In fact, the IGS1-F noncoding RNA expressed from the E-pro promoter becomes the most abundant poly(A)+ RNA in aged wild-type cells (S2B Fig). Discovery of these transcripts validates the linear model approach, which essentially identifies genes as differentially expressed based on ERC abundance if they change less with age in the mutants than in the wild type. Remaining cluster 1 genes are dominated by retrotransposable elements (23/46 genes), expression of which is known to increase in *rad52Δ* and decrease in *spt3Δ* [60,61]. These follow the expected genotype-specific behaviour starting at log phase, which makes it unlikely that retrotransposable elements are bona fide ERC-induced transcripts.

In contrast, cluster 2 is largely composed of single-copy protein coding genes, functionally enriched for negative regulation of ATPase activity, negative regulation of hydrolase activity, and cell wall organisation. However, the following lines of reasoning lead us to conclude that these genes and ontologies are not related to ERC accumulation. Firstly, the difference in expression of these genes between wild type and the mutants is highest at log phase and decreases with age (Fig 2C); this means that the expression differences are most substantial prior to ERC accumulation and therefore must be attributed to overlapping effects of the mutant genotypes rather than ERC accumulation. Indeed, 27% [30] of these genes were among the 93 mRNAs identified above as significantly more abundant in all 3 mutants at log phase. Secondly, the median model coefficient for the effect of ERC abundance on cluster 2 genes is similar to the model coefficients for Age and Genotype (S2C Fig), indicating that ERCs are, at most, a mid-ranking contributor to the differential expression of these genes.

Heterochromatic regions silenced by Sir2 followed the behaviour of ERC-regulated transcripts (rDNA and mating type loci are in cluster 1), so we asked whether titration of Sir2 by ERCs can explain the reported age-linked induction of Sir2-regulated protein coding genes [33]. Of the 40 genes repressed by the SIR complex [62], of which Sir2 is the active deacetylase, most are induced with age irrespective of ERCs as induction is similar to wild type in *rad52Δ* or *sac3Δ* (Fig 2D). The model coefficients for "high" ERC abundance and 48-hour Age for these genes, interpretable as the $\log_2$ expression change attributable to ERC accumulation and to age, were 0.7 ± 1.3 and 1.8 ± 1.7, respectively (median ± interquartile range) (S2D Fig). By comparison, these values for the cluster 2 genes were 1.7 ± 1.4 (for "high" ERC abundance) and 1.5 ± 1.5 (for 48-hour Age). Therefore, our data show that ERC accumulation may contribute to the derepression of some Sir2-regulated genes, but to a lesser extent than the derepression of the same genes that occurs due to increasing age.

Overall, despite the vast accumulation of ERCs during ageing and the resultant massive production of polyadenylated noncoding RNA, we could not identify protein coding genes for which the change in mRNA abundance during ageing is primarily driven by ERC accumulation.

## ERC accumulation is not the only pathway to the SEP

ERC accumulation immediately precedes the SEP, which is marked by formation of very bright foci of the mitochondrial outer membrane protein Tom70-GFP in G1 [5,20]. Yeast cells also undergo volume growth after entering the SEP to a point that cytoplasmic dilution could induce dysfunction [21], but it is unclear whether ERCs contribute to this growth. Both cell size and Tom70 intensity can be quantified by imaging, so we set up a high-throughput assay using the Amnis ImageStream flow cytometry system to image hundreds of individual MEP cells from purified aged MEP populations [22].

Fluorescence images of Tom70-GFP were captured to assess onset of the SEP, with ribosome component Rpl13a-mCherry as a control that does not accumulate substantially or form foci during ageing in wild type [7]. Brightfield and wheat germ agglutinin (WGA)-stained images were also acquired for each cell; cell area is calculated from brightfield, while WGA intensity forms a quantitative marker of replicative age that correlates with manual bud scar counts in wild type and *spt3Δ*, with a slight underestimate in *rad52Δ* (see comparison with manual bud scar counts in S3A Fig) [14,63,64]. As expected, in wild-type cells, Tom70-GFP and WGA intensity increased markedly from log phase to 24 hours and from 24 to 48 hours (Fig 3A and 3B). Cell size increased particularly during the first 24 hours of ageing (Fig 3C), while Rpl13a-mCherry increased only slightly (<1.5-fold, S3B Fig).

In *spt3Δ*, Tom70-GFP intensity increased with age to be significantly, although not substantially, lower than wild type at 24 hours but did not increase from 24 to 48 hours despite cells reaching equivalent age to wild type (Fig 3A and 3B). *spt3Δ* cells were slightly smaller at all ages but still grew substantially with age, while Rpl13a-mCherry intensity decreased with age, though again the change was small (Figs 3B and S3C). Cells lacking Sac3 aged more slowly than wild type, reaching an average age in 48 hours only slightly higher than wild-type cells in 24 hours, but taking this into account behaved in a very similar manner to the *spt3Δ* mutant (S3C Fig). In contrast, *rad52Δ* cells showed significantly higher Tom70-GFP fluorescence than wild type at 24 hours, even though *rad52Δ* cells were younger, as well as being slightly, though not significantly, larger (Fig 3A and 3B). None of these parameters changed from 24 to 48 hours consistent with *rad52Δ* cells reaching the end of replicative life before 24 hours, such that by 48 hours, wild-type cells had caught up in Tom70-GFP and cell size. Taken together, we observe essentially no correlation between SEP onset or cell size and the presence or absence of ERCs across this set of mutants, with *rad52Δ* being a particular outlier.

This implies that aged cells lacking ERCs should still become senescent, defined by a profound increase in cell cycle time. This is hard to assay for *rad52Δ* and *sac3Δ* mutants due to early loss of viability and slow growth but is testable in *spt3Δ* cells, which follow similar ageing kinetics to wild type (S3A Fig). Young cells placed on YPD agar by micromanipulation rapidly divide and form colonies, but post-SEP cells cycle extremely slowly, which should delay the generation of a daughter cell capable of forming a colony. We measured the sizes of colonies produced in 24 hours from individual young and 48-hour-aged cells, including only viable cells that gave rise to a colony within 72 hours, and observed an equivalent massive reduction in colony size for both aged wild-type and aged *spt3Δ* cells compared to young cells (S3D Fig). This shows that despite accumulating 10-fold less ERCs, *spt3Δ* cells become senescent with age as evidenced by very slow cell division, consistent with the Tom70-GFP intensity data. In

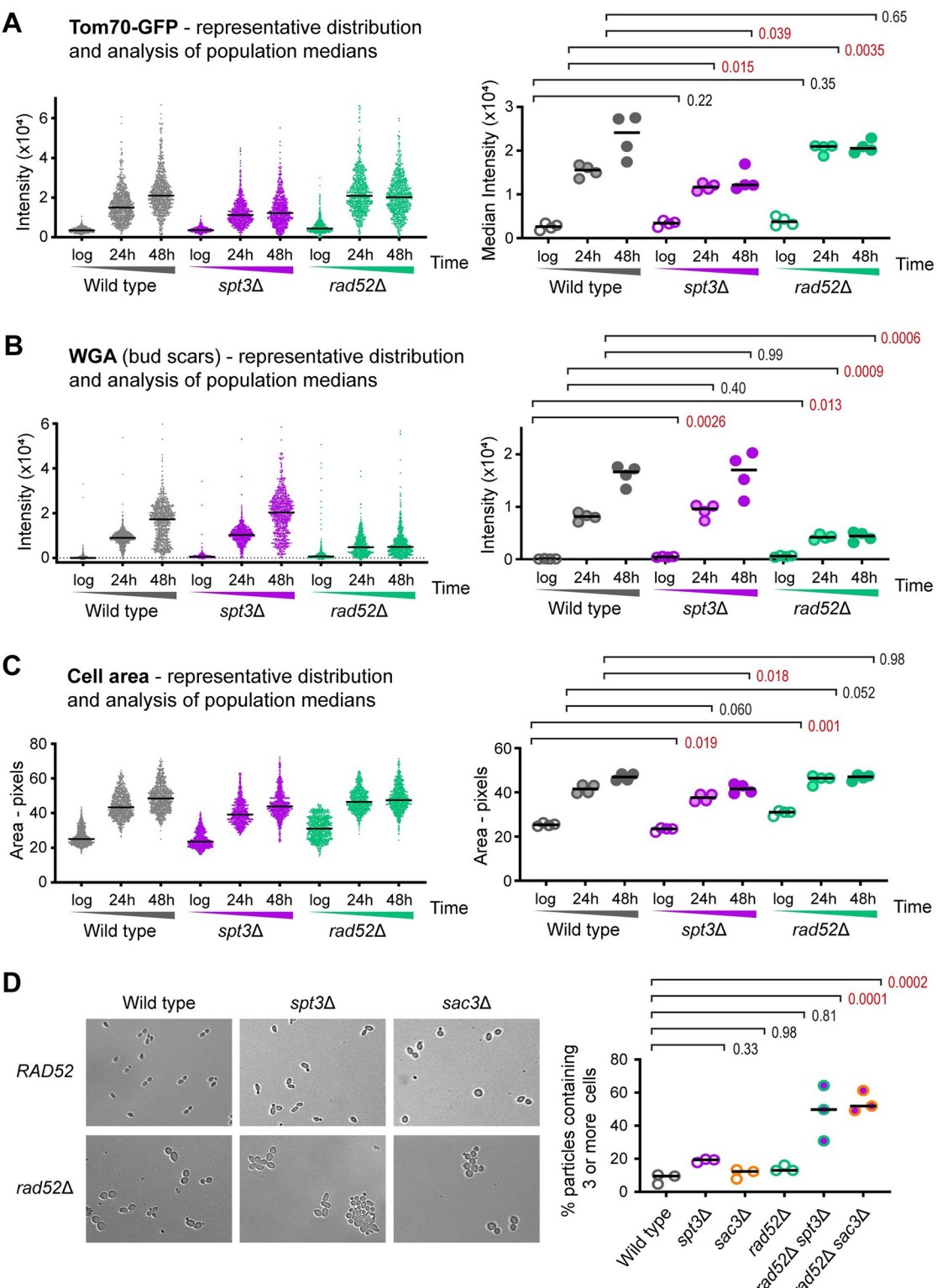

**Fig 3. Mutants lacking ERCs still display signatures of the SEP.** (**A**) Imaging flow cytometry analysis of Tom70-GFP intensity, which increases hugely in G1 cells after the SEP [5]. Cells were gated for streptavidin staining (aged samples only), which filters out young cells, then for circularity, which selects towards G1 cells and removes clumps. The left graph shows individual cells in a representative population with a bar at the median. The right graph compares medians of biological replicates. *p*-Values calculated from 2-way ANOVA with post hoc Tukey test using genotype and age as independent variables, *n* = 4. Red *p*-values indicate

significance at $p < 0.05$. (**B**) Analysis of WGA staining to show relative age. Samples and analysis as in (**A**). (**C**) Analysis of cell size in pixels extracted from brightfield images. Samples and analysis as in (**A**). (**D**) Representative brightfield images of cells from log phase cultures of indicated genotypes. Samples were fixed in ethanol and sonicated to disperse cell clumps. Quantification shows the fraction of cell particles containing 3 or more cells (a particle being a unit of 1 or more cells that is not dispersed by sonication), analysis by 1-way ANOVA with post hoc Tukey test, $n = 3$. The numerical data underlying this Figure can be found in S8 File. ERC, extrachromosomal ribosomal DNA circle; SEP, senescence entry point; WGA, wheat germ agglutinin.

contrast, wild-type cells aged on galactose accumulate high levels of ERCs but show no delay in colony formation consistent with their very low Tom70-GFP signal [22], further separating ERC accumulation from senescence.

To confirm that ageing phenotypes are not connected with ERC accumulation, we examined other previously described markers. Firstly, the size of the nucleolus, visualised as the area occupied by the RNA polymerase I marker Rpa190-GFP, has been shown to expand as cells age [20]. This is thought to result from increased rDNA copies arising through ERC amplification occupying more space and recruiting more ribosome biogenesis factors; however, we observe that this increase of nucleolar size with age occurs in wild type and in both *rad52Δ* and *spt3Δ* mutants that accumulate ≥10-fold less ERCs (S3E Fig). Secondly, the vacuole expands with age and increases in pH, resulting in a higher signal from a Vph1-mCherry reporter [65]. Again, we observe the same phenotype in wild-type and mutant cells, except that the increase stops at 24 hours in the *rad52Δ* mutant as these cells have already completed their replicative life span (S3F Fig). Our imaging data for Tom70-GFP, Vph1-mCherry, and Rpa190-GFP, as well as direct measurements of growth rate and gene expression dysregulation all lead to the same conclusion: that age-associated changes and the onset of senescence marked by the SEP occur independent of ERCs.

We expanded on these surprising observations using epistasis assays, in which *rad52Δ* was combined with *spt3Δ* or *sac3Δ*. We expected that the high Tom70-GFP and low WGA phenotypes of the *rad52Δ* mutant would be dominant in these double mutants, as it seemed likely that whatever persistent DNA damage drives the SEP in *rad52Δ* cells should occur irrespective of SAGA or TREX-2 disruption. However, we actually observed a strong additive effect. One terminal phenotype of replicatively aged yeast, particularly in diploid MEP cells, is a cell division defect that causes <5% of cells to form large multicellular aggregates (S3G Fig). However, in the *rad52Δ spt3Δ* and *rad52Δ sac3Δ* double mutants, every mother cell formed such an aggregate within 24 hours, making purification impossible. This phenotype was prominent even in log phase populations (Fig 3D), and quantifying individual particles (a particle being 1 or more cells not separable by sonication) showed that the majority of double mutant cell particles were clusters of 3 or more cells, often containing 10+ cells (Fig 3D). This observation suggests that although *rad52Δ* mutants lack ERCs, *rad52Δ* mother cells may contain another toxic species that is normally retained in mother cells by SAGA/TREX-2.

Overall, phenotypic analysis of *spt3Δ* and *sac3Δ* mutants shows that ERC accumulation makes only a minor contribution to the SEP and that *rad52Δ* develops a very strong SEP phenotype at a young age despite a complete absence of ERCs. Therefore, the presence or absence of ERCs does not define the SEP. Furthermore, the unexpected outcome of the epistasis assay raises the hypothesis that another DNA species is present in *rad52Δ* mutant cells, one that is normally retained in mother cells by SAGA and TREX-2 (just like ERCs) and which causes a cytokinesis defect if communicated to daughter cells.

## A transcriptional signature of the SEP

The uncoupling of SEP markers from ERC accumulation prompted us to reanalyse our gene expression data looking for changes associated with the SEP instead of ERCs. To do this, we

used the same dataset and linear modelling approach applied for ERCs in Fig 2, but with altered parameters to test the contribution of the SEP instead of ERCs to differential expression. We rebuilt the linear model with categorical independent variables for Age, Genotype, and *TRP1* as before, but substituted the ERC abundance variable for a continuous variable representing SEP status based on Tom70-GFP (Tom70-GFP intensity median values from Figs 3A and S3C scaled linearly from 0 to 1) (Linear Model 2, S2 File).

This model discovered 187 genes significantly differentially expressed based on the SEP variable (DESeq2, BH-corrected FDR < 0.05, $\log_2$-fold change threshold ± 0.5) (Fig 4A). Importantly, the model coefficient for SEP was far higher than other model coefficients for these genes, showing that the SEP largely explains gene expression changes for these genes across the experiment with little contribution from Age or Genotype (Fig 4B). This should be contrasted with the weak effect of ERC abundance on potential ERC-induced genes (S2C Fig). GO term enrichment analysis of the gene set differentially expressed based on the SEP revealed enrichments for sporulation, specifically genes for prospore formation, spore wall assembly and septins, as well as iron transport.

We also found 290 genes significantly differentially expressed based on Age (DESeq2 48-hour versus log phase age, BH-corrected FDR < 0.05, $\log_2$-fold change threshold ± 0.5), for which Age was by far the strongest variable contributing to differential expression with little contribution from SEP or Genotype (S4A and S4B Fig). These genes are enriched for energy utilisation categories, translation, and ribosome synthesis as may be expected for ageing, given previous reports. However, the comparison of 48 hours versus log phase also compares cells that are largely terminally arrested to cells that are not, and, therefore, some fraction of these differences may be attributable to terminal arrest rather than ageing per se.

An unexpected feature of the set of genes significantly associated with the SEP was that 22% are located on the right arm of chromosome XII between the rDNA and the telomere, a region containing only 5% of annotated genes. This region, hereafter referred to as ChrXIIr, has been previously noted to be amplified during yeast ageing with unknown effect [8,16]. We therefore asked whether the expression of genes in this region was generally disrupted in aged cells and observed that average mRNA abundance for genes on ChrXIIr increased significantly with age in wild type (S4C Fig, left, S4D Fig). Although many genes are induced with age, the average mRNA abundance change of any large, random set of genes between 2 datasets should be close to zero due to the DESeq2 normalisation method applied. Indeed, an equivalent analysis for all genes not on chromosome XII shows no average change, although the spread of mRNA abundance does increase with age as expected (S4C Fig, right, S4D Fig), so the effect on ChrXIIr is exceptional. To ensure that this accumulation of ChrXIIr is not a unique feature of ageing in the diploid strains used for the MEP, we also analysed haploid MEPs by Tom70-GFP and RNA-seq and observed a similar accumulation of ChrXIIr in concert with the increasing Tom70-GFP signal (S4E and S4F Fig).

Among the mutants, increased mRNA abundance for ChrXIIr genes was less evident in *spt3Δ* and almost undetectable in *sac3Δ*, despite a similar increase in the spread of mRNA abundance (Fig 4C and 4D, compare grey to purple and orange). In contrast, average mRNA abundance for ChrXIIr genes dramatically increased in *rad52Δ* at 24 hours, significantly more so than in wild type (Fig 4C and 4D, compare grey to green). Expression of ChrXIIr genes closely follows the pattern of the SEP marker Tom70-GFP, showing association of ChrXIIr accumulation with the SEP across our dataset (Fig 3A).

Genes on ChrXIIr were not major contributors to the GO terms enriched in the 187 SEP-associated genes; both the enriched GO terms and the model coefficients were essentially identical when genes on ChrXIIr were excluded (S4G Fig). Therefore, these 187 genes can be split into 2 classes—those located on ChrXIIr and those functionally enriched for sporulation and

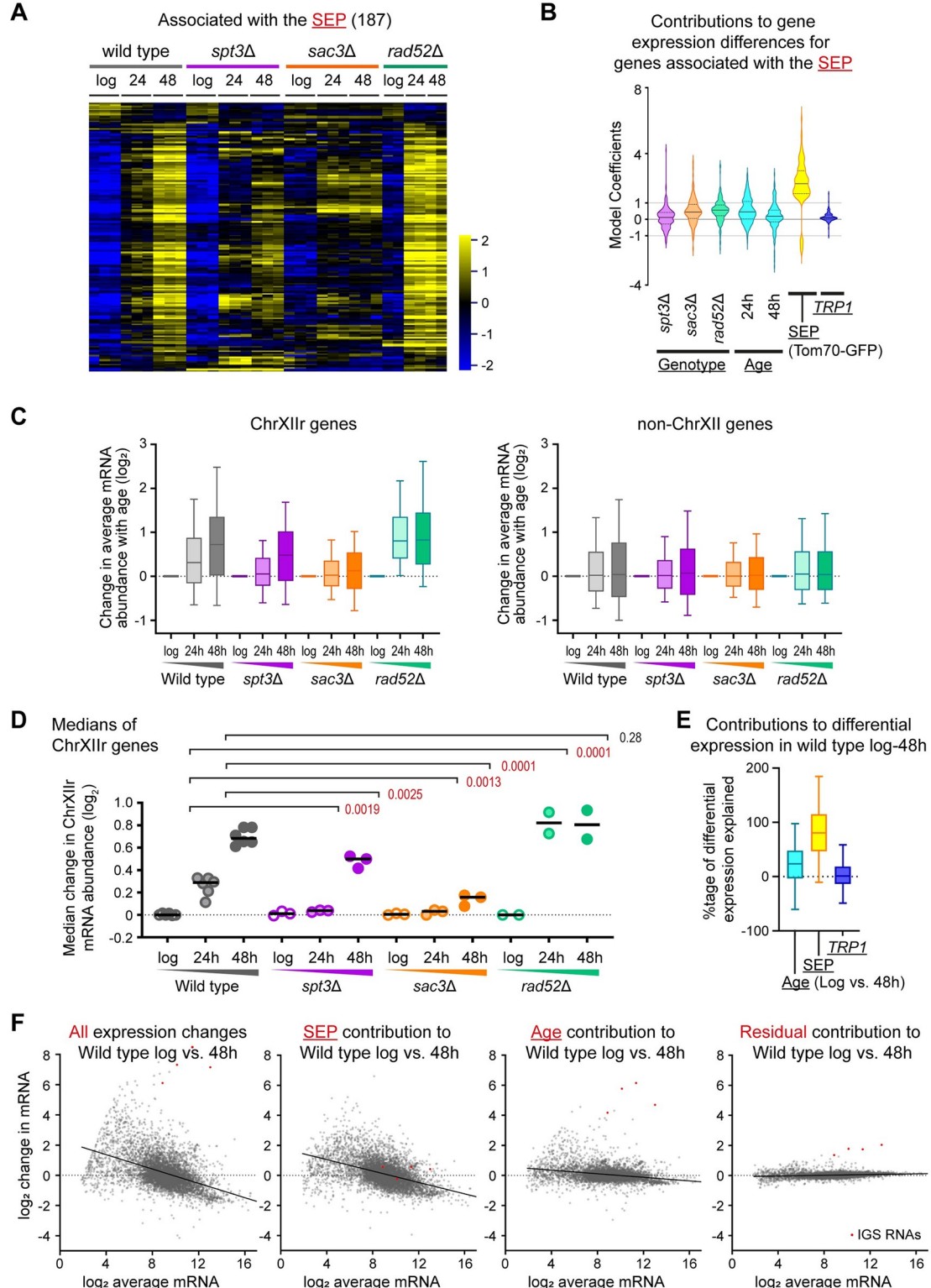

**Fig 4. A gene expression signature for the SEP.** (**A**) Hierarchical clustering of log$_2$ mRNA abundance for 187 genes called by a DESeq2 Linear Model 2 as significantly different between datasets based on the SEP (Tom70-GFP medians from Figs 3A and S3C) (DESeq2 continuous linear model factor: median Tom70-GFP scaled to between [0,1], BH-corrected FDR < 0.05, log$_2$-fold change threshold ±0.5). Individual biological replicates shown (3 for wild type, *spt3Δ*, and *sac3Δ* and 2 for *rad52Δ*. (**B**) DESeq2 Linear Model 2 coefficients representing the contribution of each individual parameter to the differences in mRNA abundance of

the set of genes shown in (**A**). DESeq2 was used to model expression as dependent on 3 categorical independent variables and 1 continuous variable (with an additional intercept factor): Genotype (wild type, *spt3Δ*, *rad52Δ*, *sac3Δ*), *TRP1* auxotrophy (present, not present), Age (log phase, 24 hours, 48 hours), and SEP (Tom70-GFP continuous, values taken from medians shown in Figs 3A and S3C min-max scaled to the range [0,1], wild type values also used for *TRP1*-repaired samples). Coefficient values can be interpreted as the log$_2$ normalised mRNA abundance change caused by a specific model factor being present relative to a wild-type log phase *TRP1* auxotroph. Solid horizontal lines within violins indicate median values, while lower and upper quartiles are indicated by dotted horizontal lines. Full output of the model is provided in S2 File. (**C**) Log$_2$ change of mRNA abundance from log phase to given time points for all genes on ChrXIIr (left) or all genes on chromosomes other than XII (right). Data are the mean of 3 biological replicates for wild type, *spt3Δ*, and *sac3Δ* or 2 biological replicates for *rad52Δ*, plots as in (**C**). (**D**) Analysis of median change in log$_2$ mRNA abundance from log phase to given time points for all genes on ChrXIIr. Analysis by 2-way ANOVA with post hoc Tukey test for aged time-points only since log values are the reference point $n = 6$ biological replicates (wild type, combining wild type and *TRP1* datasets), $n = 3$ for *spt3Δ* and *sac3Δ*, $n = 2$ for *rad52Δ*. Red *p*-values indicate significance at $p < 0.05$. (**E**) Percentage of the observed differential expression attributable to Age (log vs. 48 hours), SEP (Tom70-GFP), and *TRP1*. DESeq2 Linear Model 3 with categorical independent variables for Genotype, Age, *TRP1*, and carbon source, and 1 continuous variable for Tom70-GFP representing the SEP was applied to a dataset featuring 9 replicate time courses of wild-type cells aged on glucose (including 3 *trp1Δ*) and 5 replicate time courses of wild-type cells aged on galactose as well as the datasets for *spt3Δ*, *sac3Δ*, and *rad52Δ*. After excluding genes with a normalised read count <4 at log phase, percentages were calculated as the contribution from the given factor / the observed change × 100. Values outside the range 0%–100% represent model coefficients predicting gene expression in the wrong direction or of too great a magnitude, respectively. Full output of the model is provided in S3 File. (**F**) Assignment of gene expression change in wild type ageing log to 48 hours to SEP and Age contributions calculated by Linear Model 3, displayed as MA plots. Left–right: Observed gene expression change in wild type during ageing for average of 6 *TRP1*-positive wild-type samples at log phase and 48 hours; log phase expression with the SEP contribution added; log phase expression with the Age contribution added; log phase expression with the Residual contribution added, where the residual is observed gene expression change with age that was not attributed to SEP or Age by the model. Red dots indicate RNA from the IGS regions of the rDNA, which form prominent outliers. The numerical data underlying this Figure can be found in S8 File. BH, Benjamini–Hochberg; FDR, false discovery rate; SEP, senescence entry point.

iron metabolism. However, this gene set must only contain the strongest SEP-associated genes as age-linked dysregulation of gene expression and the SEP occur in all the mutants in the dataset to a greater or lesser extent, such that the linear model would have difficulty separating effects arising from the SEP and Age. We suspected that senescence may be responsible for the majority of reported age-linked gene expression changes but required data for cells that age without a detectable SEP phenotype to confirm this. We therefore added 5 replicate time courses for YDH18 yeast (which carries 2 fluorescent markers but is otherwise a wild-type MEP strain) aged on galactose to the dataset along with matched glucose controls; on galactose, wild-type MEP cells achieve equivalent age but with very low senescence [22]. We then recalculated the linear model with variables for SEP, Age, Genotype, and *TRP1*, and an additional variable for Carbon source (glucose or galactose). This discovered a much larger set of 1,236 genes significantly differentially expressed based on the SEP (DESeq2, BH-corrected FDR < 0.05, log$_2$-fold change threshold ± 0.5), compared to 237 based on Age, while 657 genes were significantly differentially expressed on galactose compared to glucose (Linear Model 3, S3 File).

In this variant of the model, only SEP, Age (log versus 48 hours) and *TRP1* components contribute to differential gene expression during wild type ageing on glucose. Comparing 48-hour-aged samples to log phase, the model attributes 81% of observed gene expression change to the SEP, while Age explains 24% and *TRP1* explains 1% (Fig 4E). Adding the contribution of the SEP to log phase wild-type gene expression results in a gene expression pattern very similar to that observed in aged cells including the characteristic induction of repressed genes and decreased expression of highly expressed genes, particularly translation and ribosome synthesis factors (Fig 4F, compare left-hand 2 panels). In contrast, the contribution of Age barely changes the overall shape of the gene expression profile; genes significantly affected by Age are particularly enriched for cell wall organisation genes, which may result from the cell growth that occurs in all the tested mutants as well as in galactose-aged wild-type cells and is therefore independent of the SEP (Fig 4F, Age). The biggest changes in this set are the IGS

ncRNAs transcribed from ERCs, as expected given the previous data showing no association between ERC accumulation and the SEP (Fig 4F, red points). Once Age and SEP are removed, only small fractions of the gene expression change during ageing from log to 48 hours remain unaccounted for (Fig 4F, Residual). Importantly, this bioinformatic separation of the impacts of the SEP and age itself paralleled the results of physical separation of younger, highly senescent and old, minimally senescent populations, which also showed the SEP to have a much larger effect than age itself (Figs 4 and S4 in [22]).

The SEP causes a massive slowing in growth that is predicted to induce an ESR-like gene expression change [5,13,66,67]. To confirm this, we compared the Age and SEP coefficients of gene expression during ageing to the change in expression of the approximately 900 ESR genes during a representative stress (20 minutes heat shock) reported in [13]. As predicted, a positive correlation was observed with the SEP ($R^2 = 0.68$) but no correlation with Age ($R^2 = 0.19$), showing that gene expression changes in the SEP are similar to those induced by heat shock (S5A Fig). We performed an equivalent comparison to gene expression changes that occur in response to slowing of growth [66] and also observed a correlation to the SEP (S5B Fig); the correlation was more modest ($R^2 = 0.52$), but it should be noted that culture conditions used to control growth rate were very different from those used in our ageing experiments. To generalise these associations, we compared the Age and SEP coefficients to all the stress conditions tested in [13] and observed that the correlation was much higher for the SEP than for Age in all stresses, with the best correlation to the SEP being observed for "heat shock" and "entry to stationary phase" conditions that gave the strongest ESR response in the original study. These findings demonstrate that the gene expression impact of the SEP is similar to slow growth and the ESR and should therefore largely be attributed to slowed growth. However, the SEP does not exactly equate to an ESR since expression of the 187 SEP-associated genes discovered by Linear Model 2 (Fig 4A) does not change in the same way during heat shock as during ageing (S5D Fig), and, similarly, there is no correlation between the expression change of the 187 SEP-associated genes during ageing and during change in growth rate (S5E Fig).

Overall, we find that the SEP is by far the greatest contributor to age-linked gene expression changes. The bulk of these changes result from slowing of growth, but we also find signatures of specific gene expression programmes and chromosomal abnormalities.

## ChrXIIr rather than ERC accumulation is associated with the SEP

The increased expression of genes on ChrXIIr with the SEP strongly suggested the accumulation of a chromosomal fragment from this region; indeed, a discrete chromosomal fragment arising through rDNA cleavage and containing part of the rDNA along with the chromosomal region centromere distal to the rDNA has been previously described in aged wild-type MEP cells and validated by pulsed field gel electrophoresis [8,16]. We therefore performed genome resequencing on DNA isolated from log, 24-hour, and 48 hour-aged wild type, *spt3Δ*, *rad52Δ*, and *sac3Δ* cells. The most dramatic change was rDNA copy number, which increased 14× with age in wild type due to ERC accumulation but not in the mutants (Fig 5A); however, in *rad52Δ* cells, the primary change with age was a 1.5-fold increase in ChrXIIr signal, equivalent to 1 additional copy of this region in these diploid cells (Fig 5B and 5C, left). ChrXIIr also accumulated in wild-type cells with age, but less so in *spt3Δ* and was almost undetectable in *sac3Δ* following closely the pattern we observed for SEP markers (Fig 4C), while other chromosomes did not change on average (Fig 5C, middle), and we observed little change from the rest of chromosome XII (from the left telomere to the rDNA; Fig 5C, right).

Although ChrXIIr is the most highly amplified region, other individual chromosomes also change with age albeit to a much lesser extent than ChrXIIr. We observed amplification of

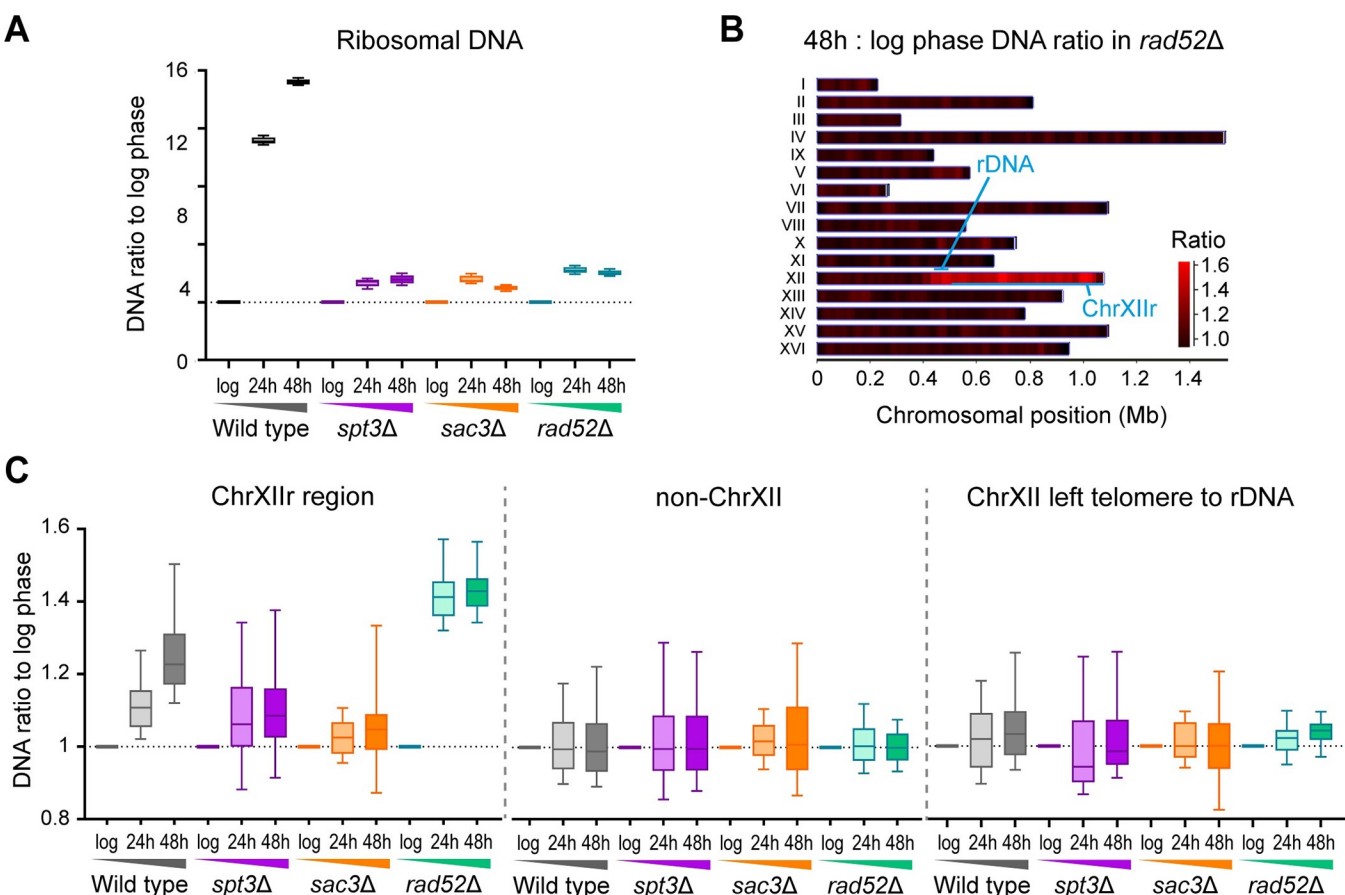

**Fig 5. ChrXIIr amplification is associated with the SEP.** (**A**) Ratio of genome resequencing read counts at given time point to log phase in 5 kb windows across the rDNA. (**B**) Heat map of read count ratios between log phase and 48 hours in *rad52Δ* in 25 kb windows across the genome. (**C**) Ratio of genome resequencing read counts at given time point to log phase in 25 kb windows across ChrXIIr (left) and non-chromosome XII regions (right). The numerical data underlying this Figure can be found in S8 File. rDNA, ribosomal DNA; SEP, senescence entry point.

chromosome I and chromosome VI by both genome resequencing and RNA-seq; although these effects are small compared to ChrXIIr, we are confident the events are real given detection in different samples by different methods (S6 Fig), while other chromosomes did not show consistent changes (S4 File). These amplification events occur to a similar extent in wild type and mutants, so are unlikely to be associated with the SEP, and are probably the outcome of chromosome missegregation events previously visualised by Neurohr and colleagues using a chromosome I marker [19]; the bias towards particular chromosomes (the shortest in the genome) is interesting but not entirely unexpected as small chromosomes in yeast are more prone to aneuploidy [68].

These experiments show that increased gene expression from ChrXIIr results from amplification of this chromosomal region, which appears particularly susceptible to aneuploidy beyond the known chromosome missegregation phenotypes observed in ageing cells. Furthermore, amplification of ChrXIIr is robustly associated with the extent of senescence across wild type, *spt3Δ*, *rad52Δ*, and *sac3Δ* mutants.

## ChrXIIr is associated with the SEP across different mutants and environments

We then tested the robustness of the association between ChrXIIr and the SEP in other mutants. *rad52Δ* is an outlier in our dataset, showing high SEP, high ChrXIIr, and no ERCs,

whereas *spt3Δ* and *sac3Δ* follow a pattern of having low ERCs and low SEP, so it was important to ensure that senescence was unconnected to ERC level in other mutants. One concern is that *rad52Δ* cells arrest at a very young replicative age, usually in G2 (>80% budded at 24 hours in our samples; S7A Fig left) [25,69], whereas wild-type yeast cells typically terminally arrest in G1. Although quantitative imaging data for Tom70-GFP and other markers is filtered to include only G1 cells and should therefore be comparable between mutants, the sequencing methods are applied to whole populations, so the unusually high budded fraction may distort the outcome. We therefore looked for further mutants and conditions that separate ERC accumulation from the SEP.

Firstly, like *rad52Δ*, the histone deacetylase double-mutant *hst3Δ hst4Δ* is short lived and reaches terminal arrest by 24 hours [70] (S7B Fig, WGA). However, a smaller fraction of cells arrest in G2/S, making the cell cycle distribution of bulk samples closer to wild-type at 24 hours (60% budded for *hst3Δ hst4Δ* compared to 80% for *rad52Δ* and 40% for wild type; S7A Fig right), and ERC levels, although reduced, are much closer to wild type (S7C Fig). Nonetheless, Tom70-GFP is substantially higher than wild type at 24 hours, showing a strong early senescence phenotype (S7B Fig, Tom70-GFP), and RNA-seq at 24 hours shows a dramatic dysregulation of gene expression compared to wild type, even more so than for *rad52Δ* (S7D Fig). Induction of the 187 genes associated with the SEP is equivalent between *hst3Δ hst4Δ* and *rad52Δ* (S7E Fig), as is the increased expression of ChrXIIr genes, indicating an equivalent ChrXIIr amplification (S7F Fig). Therefore, SEP onset, ChrXIIr accumulation, and SEP-associated gene expression changes are very similar between *hst3Δ hst4Δ* and *rad52Δ*, whereas G2 arrest and ERC accumulation are different, meaning that the rapid, strong SEP in short-lived mutants is strongly associated with ChrXIIr accumulation and cannot be attributed to G2-biased cell cycle distribution at terminal arrest or to a peculiarity of the *rad52Δ* mutant.

Secondly, cells lacking the endonuclease Mus81 accumulate 10-fold fewer ERCs (Fig 6A and [71]). However, *mus81Δ* cells do not terminally arrest at a young age, reaching equivalent age to wild type by 24 hours and dividing further from 24 to 48 hours, even though age at 48 hours is less than wild type (Fig 6B, WGA). Furthermore, *mus81Δ* cells arrest less often in G2/S compared to *rad52Δ* cells (60% budded for *mus81Δ* compared to 80% for *rad52Δ*; Fig 6C). However, despite the lack of ERCs, *mus81Δ* cells accumulate similar levels of Tom70-GFP to wild type (Fig 6B, Tom70-GFP) and undergo equivalent gene expression dysregulation (S7G Fig). ChrXIIr gene expression rises similarly to wild type in keeping with Tom70-GFP intensity (Fig 6D), and induction of SEP-associated genes is also equivalent to wild type (Fig 6E). Altogether, this demonstrates that *mus81Δ* cells show a normal SEP profile despite a large reduction in ERC accumulation, without the strong and early induction of the SEP observed in short-lived *rad52Δ* an *hst3Δ hst4Δ* mutants, and that the SEP is again tightly associated with ChrXIIr amplification.

Thirdly, to avoid peculiarities of DNA repair mutants, we asked whether ChrXIIr and/or ERC accumulation predict the SEP marker Tom70-GFP in wild-type cells aged in different conditions using the large dataset of matched RNA-seq and Tom70-GFP data from [22]. A plot of ChrXIIr accumulation versus Tom70-GFP across all datasets reveals a very good correlation ($R^2 = 0.84$) (Fig 6F), showing that ChrXIIr is tightly associated with the SEP. The same dataset can also be used to examine ERCs as the change in reverse strand signal from ncRNA in the IGS2 region efficiently reports ERC accumulation (compare S2A with S1B Fig; also compare Fig 5A and 5B in [22]), providing a proxy for ERC accumulation in RNA-seq data. As expected, given our RNA-seq analysis, we observe less correlation between ERC accumulation and Tom70-GFP ($R^2 = 0.47$), and closer inspection reveals that the data are subdivided in 2 sets—a low Tom70-GFP set with a wide spectrum of ERC levels, and a high IGS expression set with a wide spectrum of Tom-70 GFP signal (Fig 6G, red and blue, respectively)—in each of which there is essentially no correlation between ERCs and Tom70-GFP.

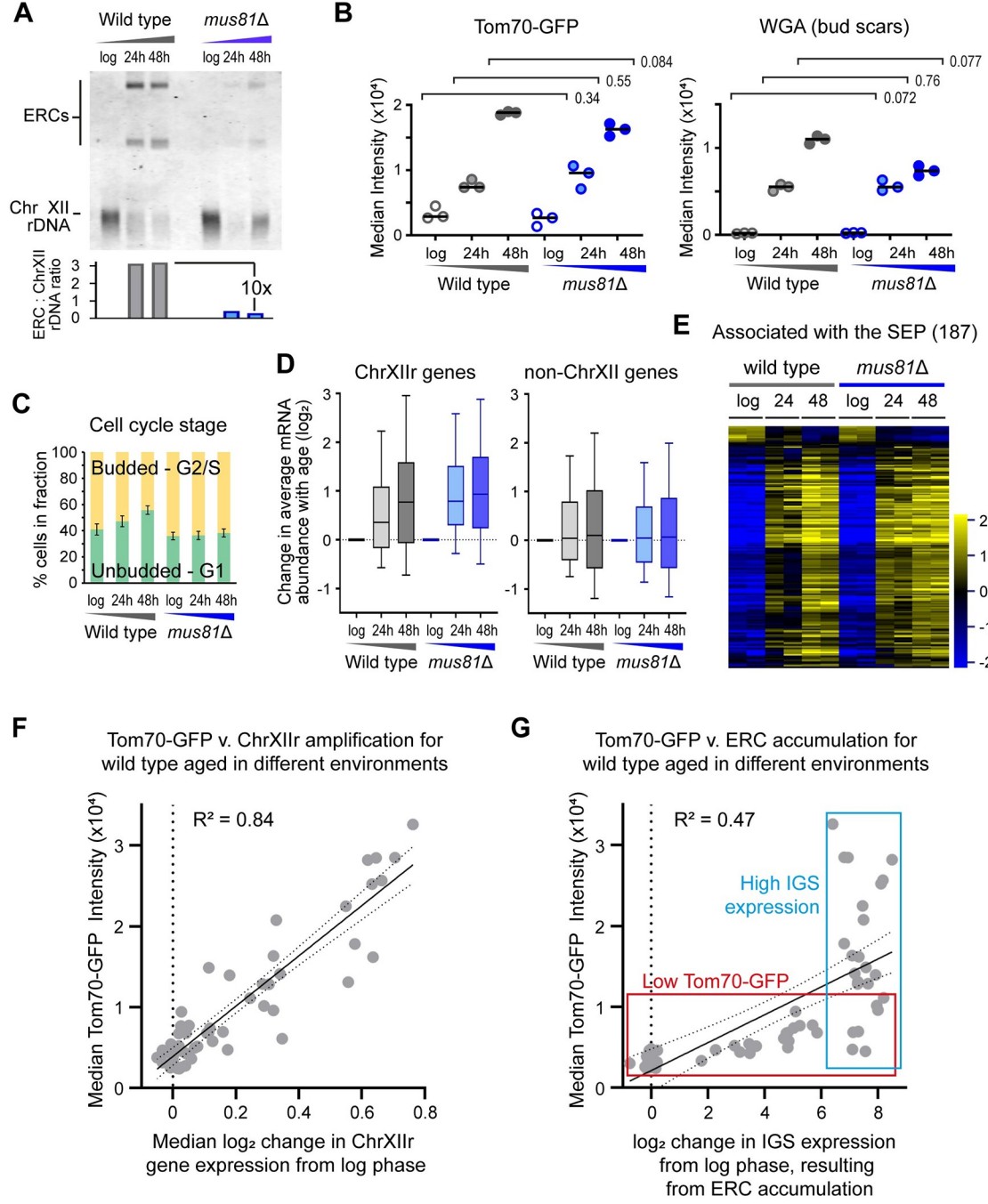

**Fig 6. Robust association of ChrXIIr amplification with the SEP.** (**A**) Southern blot analysis of ERC abundance in log phase and 48-hour-aged wild-type and *mus81Δ* MEP cells; quantification shows ratio of ERC bands to chromosomal rDNA. (**B**) Imaging flow cytometry analysis of Tom70-GFP intensity, which increases hugely in G1 cells after the SEP [5], and for WGA, which reports on replicative age. Cells were gated for streptavidin staining (aged samples only), which filters out young cells, then for circularity, which selects towards G1 cells and removes clumps. Graph shows medians of biological replicate populations, *p*-values calculated from 2-way ANOVA with post hoc Tukey test using genotype and age as independent variables, *n* = 3. Red *p*-values indicate significance at $p < 0.05$. (**C**) Cell cycle stage plot comparing the percentage of budded and unbudded cells. Imaging flow cytometry was performed with gating for streptavidin (aged samples only), then budded and unbudded cells called based on circularity and validated by examining representative images. Error bars ±1 SD. (**D**) $\log_2$ change of mRNA abundance from log phase to given time points for all genes on ChrXIIr (left) or all genes on chromosomes other than XII (right). Data are the mean of 3 biological replicates. (**E**) Hierarchical clustering of $\log_2$ mRNA abundance for 187 genes called by a DESeq2 Linear Model 2 as significantly different between datasets based on the SEP. Individual biological replicates shown (2 for each condition). (**F**) Plot of senescence based on Tom70-GFP versus ChrXIIr amplification (estimated from RNA-seq), for all individual wild-type samples from [22], for

which matched Tom70-GFP and gene expression data are available (GSE207503 glucose, galactose, fructose, sucrose, raffinose-galactose, glucose<>galactose shifts). $R^2$ calculated by linear regression; dotted lines show 95% confidence intervals. (**G**) Plot of senescence based on Tom70-GFP versus ERC accumulation (estimated from IGS2 signal at given time point–log in RNA-seq), for all individual wild-type samples from [22], for which matched Tom70-GFP and gene expression data are available (GSE207503 glucose, galactose, fructose, sucrose, raffinose-galactose, glucose<>galactose shifts). $R^2$ calculated by linear regression; dotted lines show 95% confidence intervals. Red and blue boxes outline the 2 observable clusters of points. The numerical data underlying this Figure can be found in S8 File. ERC, extrachromosomal ribosomal DNA circle; MEP, mother enrichment program; rDNA, ribosomal DNA; SEP, senescence entry point; WGA, wheat germ agglutinin.

Finally, the particularly strong SEP phenotype of the *rad52Δ* mutant, as well as its profound DNA repair deficit, could result in the *rad52Δ* dataset having an undue influence in our linear models wherein we find an association between the SEP and the ChrXIIr chromosomal abnormality. To address this concern, we refitted Linear Models 2 and 3 including data for wild type, *spt3Δ*, and *sac3Δ* but without the *rad52Δ* dataset (S5 and S6 Files). Reassuringly, most genes significantly differentially expressed based on the SEP variable in the original models were still significantly differentially expressed when the *rad52Δ* data were excluded (78% of genes in Model 2 and 90% of genes in Model 3). Furthermore, genes significantly differentially expressed based on the SEP were still strongly overrepresented on ChrXIIr but not on other chromosomes when *rad52Δ* was excluded from the analysis, showing that the association of ChrXIIr with the SEP is maintained (S8A Fig). Quantitatively, individual coefficients from Model 3 fitted without the *rad52Δ* dataset were very similar to the original model (median difference in Age coefficient 0.02 ± 0.03 SD, SEP coefficient 0.04 ± 0.09 SD), and, unsurprisingly, plots of these components are almost identical to those from the model fitted with the full dataset (compare S8B Fig to Fig 4F). Although the component of IGS transcription attributed to the SEP by the model lacking the *rad52Δ* dataset is higher than in the original model, this is still minor compared to the component associated with Age; this means that ERC amplification is still primarily associated with age rather than SEP onset (S8B Fig, red dots). These tests clearly demonstrate that our findings are not dependent on the dataset from the *rad52Δ* mutant.

ChrXIIr amplification is therefore robustly associated with the SEP across many mutants and environments, whereas ERCs are not, showing that ChrXIIr accumulation is a major feature of senescence.

## Amplification of protein coding genes on ChrXIIr does not cause senescence

ChrXIIr can be imagined in 2 parts—a tract of rDNA repeats of variable size from 0 to 1.8 Mb depending on where within the rDNA the cleavage occurs that forms ChrXIIr, attached to a region of 0.6 Mb containing 334 annotated genes (Fig 7A, schematic). We considered 2 potential mechanisms by which accumulation of such a noncentromeric chromosome fragment could contribute to the SEP. The dysregulation of expression of the 334 genes caused by the imbalance in gene copy number could result in widespread defects in protein complement [72], or alternatively, the presence of a large noncentromeric chromosomal fragment could itself impair the cell cycle irrespective of gene content. Of course, across the spectrum of age-linked pathology, these explanations may not be mutually exclusive.

To determine whether dysregulation of the 334 genes on ChrXIIr is associated with the SEP, we engineered a translocation that relocated the gene-containing 0.6 Mb section of ChrXIIr to chromosome V and terminated chromosome XII with a telomere adjacent to the rDNA, forming the XIIr <> V strain (Figs 7B and S9A). This should prevent the amplification of the 334 genes, but not impede the amplification of the variable length rDNA-containing

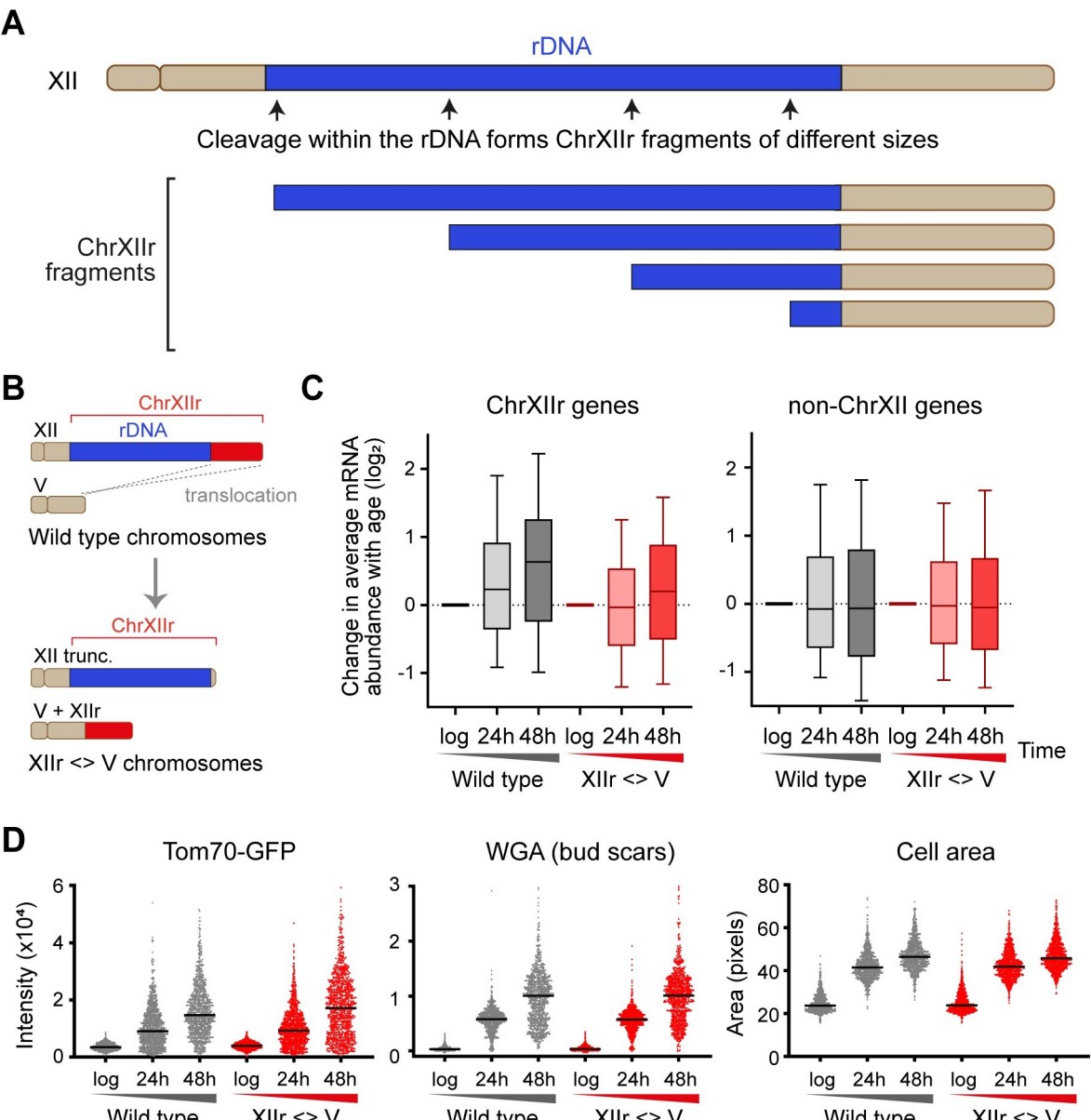

**Fig 7. Increased copy number of genes telomere proximal to the rDNA on ChrXII is not responsible for the SEP.** (**A**) Schematic showing different possible sizes of ChrXIIr and relative contributions from other regions of Chromosome XII. (**B**) Schematic of translocation strategy for fusing gene-containing portion of ChrXIIr to chromosome V. (**C**) Log$_2$ change of normalised mRNA abundance from log phase to given time points for all genes on ChrXIIr (left) or all genes on chromosomes other than XII (right). (**D**) Plots of Tom70-GFP and WGA intensity with cell size gated as in Fig 3 for wild type and chromosome XIIr <> V translocation strain. The numerical data underlying this Figure can be found in S8 File. rDNA, ribosomal DNA; SEP, senescence entry point; WGA, wheat germ agglutinin.

segment, such that a ChrXIIr-like species can still form but containing only the rDNA tract without the protein-coding genes. For reasons that remain unclear, we were unable to generate this translocation in the α haploid and therefore generated a:a diploids of both the wild-type control strain and the strain carrying the XIIr <> V translocation. The increase in ChrXIIr gene mRNAs that occurs with age in wild type was dramatically reduced in the XIIr <> V strain, which is as expected since these genes are no longer present on the ChrXIIr fragment

accumulated after rDNA cleavage (Fig 7C). Nonetheless, the increase in Tom70-GFP signal with age and the growth in cell size occurred just as in wild type (Fig 7D), while global gene expression dysregulation and change in expression of the SEP-linked genes defined in Fig 4A occurred just as in the control strain (S9B and S9C Fig). This experiment clearly shows that although the SEP is associated with amplification of ChrXIIr, the SEP does not require amplification of specific genes on ChrXIIr. In other words, amplification of centromere distal genes to the rDNA does not cause senescence; instead, this is presumably associated with the formation of the large linear rDNA component of the ChrXIIr fragment.

Overall, our results are entirely consistent with models that predict defects in rDNA stability to be potent drivers of ageing and age-linked phenotypes. However, we find no evidence that ERCs have any impact on the physiology of aged cells; rather, the formation of a noncentromeric fragment encompassing the region between the rDNA and the right telomere of chromosome XII is associated with the SEP and is a strong candidate for mediating the damaging effects of ageing in yeast.

## Discussion

ERCs were first proposed as a cause of yeast ageing over 20 years ago, based on strong accumulation in short-lived mutants and the life span reduction caused by artificial ERCs [24]. However, the mechanism of ERC toxicity remains unknown, and though recent studies linked defects in ribosome synthesis and nuclear pore structure to ERC accumulation, it is unclear whether or how these affect ageing cell physiology [19,20,49,50]. Here, we examined ageing phenotypes in a panel of low-ERC mutants and found that age-associated transcriptomic dysregulation and the SEP occur independently of ERCs. Furthermore, we did not find any robust indication of pathway-specific impacts of ERCs at the gene expression level. ERCs are therefore not necessary for age-associated phenotypes in yeast, and in the accompanying manuscript, we show that ERCs are also not sufficient as cells aged for 48 hours on galactose have plentiful ERCs but minimal age-associated phenotypes [22]. These findings are incompatible with the hypothesis that ERCs are a primary driver of age-related changes in *S. cerevisiae*. Instead, we show that another rDNA-derived extrachromosomal species is tightly associated with the SEP and may be a major contributor to age-related phenotypes in normal yeast cells.

### ChrXIIr as a candidate mediator of ageing phenotypes in yeast

Substantial effort has been invested in understanding the mechanism by which ERC accumulation can cause ageing. Given that ChrXIIr levels are much more tightly associated with the SEP than ERC levels, we asked whether these mechanisms could be adapted to explain how ChrXIIr present at approximately 1 molecule per cell might induce age-linked phenotypes. Recent studies have identified ultrastructural changes in the nuclear pore and nuclear pore complex impairment as causes of ageing phenotypes in yeast, arising through binding of ERCs to nuclear pores [49,50]. We propose that large linear acentromeric DNA fragments also bind nuclear pore complexes by the same SAGA/TREX-2-mediated mechanism as ERCs but due to their size would link together and entangle many nuclear pore complexes, thereby impairing nuclear membrane structure (Fig 8). Interestingly, this recapitulates the pathogenic mechanism that underlies accelerated ageing in HGPS, which involves disruption of nuclear membrane architecture, linking ageing mechanisms in yeast and humans [73].

This mechanism should be largely independent of the sequence composition of the DNA fragments, being simply driven by large fragment size. This explains why translocation of the 600-kb centromere distal region from Chromosome XII did not prevent the SEP as rDNA cleavage events would still form linear acentromeric DNA fragments of up to 1.8 Mb (Fig 7C),

and the defects caused per molecule would not be much different. Others have shown that synthetic ERCs can cause life span and nuclear pore defects, although rDNA sequences themselves are dispensable [24,49,74]; it is likely that small circular DNAs the size of a single ERC could entangle a few nuclear pores, causing a pathogenic phenotype if present at sufficient levels, but the model we propose predicts that these would be far less disruptive than a much smaller number of very large linear molecules such as ChrXIIr. It remains unknown why ERCs bind to nuclear pore complexes, whereas chromosomal rDNA repeats seemingly do not, though one possibility is that this interaction occurs for all rDNA copies but is disrupted by chromosome condensation, which requires a centromere [75]. As ERCs and ChrXIIr lack centromeres, both species would remain bound to nuclear pore complexes into mitosis, whereas full-length chromosome XII would not; in support of this idea, condensin mutants show a specific retention of chromosome XII in the mother cell along with ERCs [76].

Defects in nuclear pore complexes and nuclear membrane structure would be expected to cause nonspecific changes in mRNA export and gene expression, rather than impacting particular sets of genes, and such changes would be invisible to bulk RNA-seq methods. Dysregulation of gene expression would impair the precise biochemical control that facilitates rapid growth, slowing the cell cycle, which, in turn, would cause a change in the global expression profile as cells alter gene expression suitably for the reduced growth rate. This explains our finding that many of the gene expression changes previously attributed to replicative ageing are more specifically a consequence of the SEP, with post-SEP cells displaying a highly dysregulated gene expression pattern that resembles the effect of slow growth.

It seems reasonable to suggest that prolonged time in the SEP with high levels of nuclear pore complex and nuclear membrane disruption will eventually render cells inviable, either due to gene expression pattern being disrupted beyond tolerable parameters or due to known impacts of nuclear membrane structure defects on DNA replication (Fig 8). Either effect could explain the early terminal arrest of *rad52Δ* and *hst3Δ hst4Δ* cells. In general, however, our data do not support the SEP being the primary life-limiting process in normal yeast ageing: Nonsenescent galactose-aged cells have a similar life span to cells aged on glucose which do senesce; nonsenescent *sac3Δ* cells are short lived; and Li and colleagues have described a SEP-free ageing trajectory by microfluidics [23]. Therefore, ChrXIIr is tightly associated with senescence and poorly with life span, making ChrXIIr a candidate mediator of age-linked phenotypes but not a life span limiter except perhaps in extreme cases.

The ERC model of yeast ageing has been remarkably effective in predicting and explaining yeast mutants with altered life spans. However, these outcomes are difficult to translate to higher eukaryotes where evidence for ERC accumulation in ageing is lacking. Our findings suggest that other, lower copy number extrachromosomal DNA species could be potent mediators of ageing phenotypes, driving differences in ageing health almost independent of life span. Of course, it is not yet proven that the appearance of ChrXIIr always triggers SEP onset, so we cannot rule out the possibility that ChrXIIr formation is a consequence rather than a cause of the SEP. However, non-centromeric fragments have the potential to mediate long-term phenotypic effects of ageing and could arise through age-linked rDNA cleavage in higher eukaryotes, raising the prospect of an rDNA stability mechanism of ageing pathology conserved from yeast to mammals.

## ChrXIIr accumulation can substitute for ERCs in many models of ageing

The accumulation of ChrXIIr first reported by Hu and colleagues is likely to be affected by the vast majority of the mutants used to modulate ERC levels, though mechanistic dependencies for ChrXIIr and ERCs are not identical. The relationship between ERCs and ageing has been

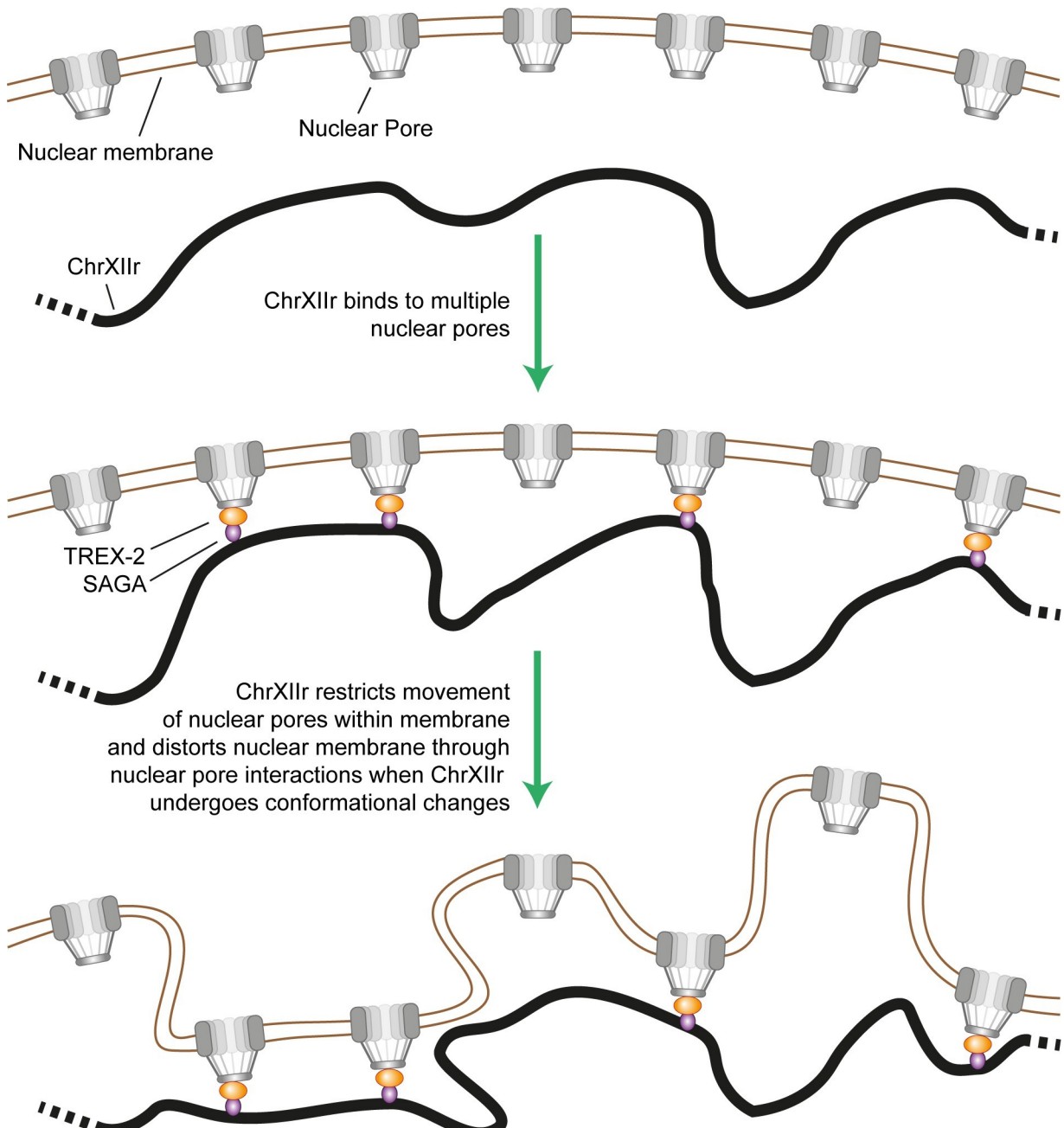

**Fig 8. Hypothesised mechanism for pathological activity of ChrXIIr.** A single copy of ChrXIIr is bound through SAGA and TREX-2 to many nuclear pore complexes. As ChrXIIr undergoes conformational changes and random motion, this will exert forces that act to deform the nuclear membrane leading to the observed defects in nuclear membrane structure [50]. This will also restrict any required redistribution and relaxation of nuclear pore complexes within the nuclear membrane, and ChrXIIr binding would be expected to induce the same gradual impairment of nuclear pore structure and function attributed to ERCs [49].

previously questioned by Ganley and colleagues who demonstrated that reducing the efficiency of rDNA replication origins not only decreased ERC abundance but also decreased life span and rDNA stability [77]; we note that irrespective of the effect on ERCs, reducing origin efficiency would increase the chance that rDNA replication is not completed in a timely fashion and accelerate ChrXIIr formation, driving an early very strong SEP. Similarly, the ChrXIIr

model can resolve the long-standing issue of why homologous recombination mutants are short lived despite having no ERCs, since ChrXIIr formation should be accelerated when homologous recombination is not available to repair stalled replication forks, likely leading to early terminal arrest through an unusually strong SEP [25].

Three recent studies have indicated that ERCs cause the SEP, which must be resolved if the ChrXIIr model is correct: Morlot and colleagues showed that the timing of ERC accumulation above background immediately precedes SEP onset [20]. However, this was only observed on average, with many individual cells showing ERC accumulation long before or long after the SEP. The risk of rDNA replication problems is known to increase with age, and we suspect that these defects are the source of both ChrXIIr and accumulated ERCs [52]. Under this model, ERC accumulation and SEP onset caused by ChrXIIr would be coincident on average because they arise through the same underlying mechanism. Neurohr and colleagues showed that the senescence phenotype can transfer to daughter cells along with all ERCs during occasional catastrophic missegregation events [19]. However, given that the mother cell retention mechanism of ChrXIIr is probably the same as for ERCs, ChrXIIr would probably have been transferred to the daughter with the ERCs. Indeed, the Cfi1 marker used in this study would not distinguish ChrXIIr from ERCs. Finally, Li and colleagues reported nucleolar expansion and fragmentation, which is thought to result from ERC accumulation [23]. However, we now show that nucleolar expansion occurs independently of ERCs (S3E Fig) so cannot be taken as evidence of a contribution of ERCs to senescence. Therefore, the similar biogenesis and segregation behaviour of ChrXIIr means that ChrXIIr could underlie many effects previously attributed to ERCs.

## A mechanism for ChrXIIr formation

Much of the evidence connecting ERCs to ageing derives from mutations that increase or decrease ERC levels in young cells. However, these mutants are not absolutely specific, and those that impact rDNA recombination will also affect the formation of other nonchromosomal DNA species, including ChrXIIr. Many studies have attributed rDNA instability and ERC formation to replication fork stalling at the Fob1-mediated RFB, and SEP onset does not occur in approximately 70% of haploid *fob1Δ* cells [20,26,31–33,78,79]. If ChrXIIr causes the SEP, then formation must be promoted by Fob1. However, an alternative Fob1-independent pathway must also exist since approximately 30% of *fob1Δ* cells also undergo the SEP accompanied by strong ERC accumulation [20]. Given that we detect no difference in ERC abundance between aged wild type and *fob1Δ*, this Fob1-independent pathway may well be dominant in diploid MEP cells, so any mechanism for ChrXIIr formation must feature Fob1 but as a contributor rather than an indispensable component. Fob1 also promotes, but is not required for, loss of heterozygosity events at ChrXIIr that increase with age in diploid MEPs [52].

Mechanistically, Fob1 blocks the passage of replication forks that would otherwise encounter the 35S RNA polymerase I transcription unit head-on (S10Aii Fig), which are then resolved by replication forks travelling codirectionally with RNA polymerase I. However, if the codirectional fork becomes unrecoverably stalled, then replication is not completed and a stable unreplicated structure would remain (S10Aiii Fig). DNA replication in yeast can continue into anaphase, at which point this structure would become stretched between 2 mitotic spindles to form a bridging chromosome (S10Aiv Fig); indeed, single-stranded rDNA intermediates have been directly observed to persist into anaphase in aged yeast [19]. DNA replication in yeast can continue into anaphase, at which point this structure would become stretched between 2 mitotic spindles to form a bridging chromosome [80–82] (S10Aiv Fig). The replication fork

structures must then be cleaved to allow completion of mitosis, as observed in mammalian systems [80,83–85], forming the ChrXIIr fragment as well as a truncated chromosome XII (S10Av Fig). Under this model, ChrXIIr forms through cleavage of replication intermediates that remain unresolved at anaphase, rather than through specific recombination events.

Replication fork stalling at the RFB does not occur in *fob1Δ*, but it is likely that head-on collisions with RNA polymerase I would also cause unrecoverable fork stalling with the same outcome at a substantial frequency. Conversely, we predict that the frequency of unrecoverable stalling of replication forks in the rDNA would increase in a *rad52Δ* mutant as homologous recombination often rescues stalled replication forks [86,87], so ChrXIIr formation would occur more often and earlier during ageing in homologous recombination-deficient mutants. This would explain why the SEP phenotype is stronger in homologous recombination mutants than in wild type despite a complete lack of ERCs.

The ChrXIIr formation mechanism can also result in loss of heterozygosity in diploid cells (S10B Fig). If the intact chromosome XII segregates to the daughter after cleavage, the 2 cleaved chromosome XII fragments would remain in the mother cell to be repaired by single-strand annealing in the next cell cycle (S10B Fig, left outcome) [88]. However, if the centromeric fragment of the cleaved chromosome XII segregates to the daughter nucleus, it becomes physically separated from the ChrXIIr fragment, which is not under tension and will be recruited to nuclear pores in the mother nucleus [89] (Fig 8). This results in loss of heterozygosity for the daughter cell and ChrXIIr accumulation in the mother cell. Therefore, loss of heterozygosity events can be mechanistically linked to ChrXIIr formation and the SEP.

In summary, although ERC accumulation does not explain entry to the SEP, formation of another rDNA-derived species ChrXIIr is robustly associated with the SEP and provides a potential mechanism for the onset of senescence during ageing in yeast. Moreover, in tandem with [22], we find that most of the gene expression changes canonically associated with yeast ageing are actually an outcome of senescence rather than replicative age per se.

## Materials and methods

Detailed and updated protocols are available at https://www.babraham.ac.uk/our-research/epigenetics/jon-houseley/protocols.

## Yeast culture and labelling

Yeast strains were constructed by standard methods and are listed in S1 Table, oligonucleotide sequences are given in S2 Table, and plasmids in S3 Table. For chromosome V <> XII translocation, Tom70-GFP was first tagged with GFP in MEP **a** using a variant of pFA6a-GFP-KanMX6 in which the I-*Sce*I site had been removed by site directed mutagenesis using oligos oJH1819-20, then *ade2Δ* was reverted by transforming with an *ADE2* fragment from BY4741 to form YJH1581. The Kan resistance gene in inducible I-*Sce*I plasmid pGSKU [90] (*Sac*I-*Bmg*BI fragment) was replaced with the *TRP1* marker from pFA6a-TRP1 [91] (*Sac*I-*Sma*I fragment) to form pGSTU. Then, a 2-kb fragment of chromosome V with an I-*Sce*I site in the middle was inserted into pGSTU, using PCRs with oligos oJH1798-9 (*Sac*I-*Bam*HI) and oJH1800-1 (*Sac*II-*Bam*HI), and *Sac*I *Sac*II digested pGSTU to form pGST-V-U. The PCR product oJH1808-9 on this plasmid was transformed into YJH1581 to give yJH1585 and also into the MEP **a** haploid to form YJH1559. Colonies of YJH1585 and YJH1559 were restreaked twice on YPGalactose plates before screening colonies by PCR for reciprocal transformations and confirmation by PFGE (0.5× TBE, 1% gel, 60- to 120-second switching, 6 V/cm, 14°C, 24 hours), forming YJH1594 and YJH1560. The NatMX6 markers in YJH1584 and YJH1594 were replaced with His3MX6, forming YJH1703 and YJH1704. YJH1703 was mated to YJH1559,

while YJH1704 was mated to YJH1560, and **a:a** diploids YJH1725 and YJH1726 were selected on YPD Kan Nat plates.

All cells were grown in YPD media (2% peptone, 1% yeast extract, 2% glucose) at 30˚C with shaking at 200 rpm. Media components were purchased from Formedium, and media was sterilised by filtration. MEP experiments were performed as described with modifications [15]: Cells were inoculated in 4 ml YPD and grown for 6 to 8 hours and then diluted in 25 ml YPD and grown for 16 to 18 hours to 0.2 to $0.6 \times 10^7$ cells/ml in 50 ml Erlenmeyer flasks. $0.125 \times 10^7$ cells per aged culture were harvested by centrifugation (15 seconds at 13,000$g$), washed twice with 125 μl PBS, and resuspended in 125 μl of PBS containing approximately 3 mg/ml Biotin-NHS (Pierce 10538723). Cells were incubated for 30 minutes on a wheel at room temperature, washed once with 125 μl PBS, and resuspended in 125 μl YPD, then inoculated in 125 ml YPD at $1 \times 10^4$ cells/ml in a 250-ml Erlenmeyer flask (FisherBrand FB33132) sealed with Parafilm. 1 μM β-estradiol (from stock of 1 mM Sigma E2758 in ethanol) was added after 2 hours for wild type and *spt3Δ*, 3 hours for *sac3Δ* and *mus81Δ*, and 4 hours for *rad52Δ* and *hst3Δ hst4Δ*. An additional $0.125 \times 10^7$ cells were harvested from the log phase culture, while biotin labelling reactions were incubating at room temperature. Cells were harvested by centrifugation for 1 minute, 4,600 rpm, immediately fixed by resuspension in 70% ethanol, and stored at −80˚C. To minimise fluorophore bleaching during Tom70-GFP experiments, the window of the incubator was covered with aluminium foil, lights on the laminar flow hood were not used during labelling, and tubes were covered with aluminium foil during biotin incubation.

## Cell purification

Percoll gradients (1 to 2 per sample depending on harvest density) were formed by vortexing 900 μl Percoll (Sigma P1644) with 100 μl 10× PBS in 2 ml tubes and centrifuging 15 minutes at 15,000$g$, 4˚C. Ethanol-fixed cells were defrosted and washed with 1 volume of cold PBSE (PBS + 2 mM EDTA) before resuspension in approximately 100 μl cold PBSE per gradient and layering on top of the preformed gradients. Gradients were centrifuged for 4 minutes at 2,000$g$, 4˚C, and then the upper phase and brown layer of cell debris were removed and discarded. 1 ml PBSE was added, mixed by inversion, and centrifuged 1 minute at 2,000$g$, 4˚C to pellet the cells, which were then resuspended in 1 ml PBSE per time point (reuniting samples that were split across 2 gradients). 25 μl Streptavidin microbeads (Miltenyi Biotech 1010007) were added and cells incubated for 5 minutes on a wheel at room temperature. Meanwhile, 1 LS column per sample (Miltenyi Biotech 1050236) was loaded on a QuadroMACS magnet and equilibrated with cold PBSE in a 4˚C room. Cells were loaded on columns and allowed to flow through under gravity, washed with 1 ml cold PBSE, and eluted with 1 ml PBSE using plunger. Cells were reloaded on the same columns after re-equilibration with approximately 500 μl PBSE, washed and reeluted, and this process repeated for a total of 3 successive purifications. After addition of Triton X-100 to 0.01% to aid pelleting, cells were split into 2 fractions in 1.5 ml tubes, pelleted 30 seconds at 20,000$g$, 4˚C, frozen on N2, and stored at −70˚C.

For live purification, cells were pelleted, washed twice with synthetic complete 2% glucose media, then resuspended in 2 ml final volume of the same media and incubated for 5 minutes on a rotating wheel with 10 μl MyOne streptavidin magnetic beads (Thermo), isolated using a magnet and washed 5 times with 1 ml of the same media. For colony formation assays, cells were streaked on a YPD plate and individual cells moved to specific locations using a Singer MSM400 micromanipulator. Colony size was measured 24 hours later on the screen of the micromanipulator imaging with a 4× objective.

## DNA extraction and Southern blot analysis

Cell pellets were resuspended in 50 μl 0.34 U/ml lyticase (Sigma L4025) in 1.2 M sorbitol, 50 mM EDTA, and 10 mM DTT and incubated at 37°C for 45 minutes. After centrifugation at 1,000*g* for 5 minutes, cells were gently resuspended in 80 μl of 0.3% SDS, 50 mM EDTA, and 250 μg/ml Proteinase K (Roche 3115801) and incubated at 65°C for 30 minutes. 32 μl 5 M KOAc was added after cooling to room temperature; samples were mixed by flicking and then chilled on ice for 30 minutes. After 10-minute centrifugation at 20,000*g*, the supernatant was extracted into a new tube using a cut tip; 125 μl phenol:chloroform (pH 8) was added, and samples were mixed on a wheel for 30 minutes. Samples were centrifuged for 5 minutes at 20,000*g*; the upper phase was extracted using cut tips and precipitated with 250 μl ethanol. Pellets were washed with 70% ethanol, air-dried, and left overnight at 4°C to dissolve in 20 μl TE. After gentle mixing, 10 μl of each sample was digested with 20 U *Xho*I (NEB) for 3 to 6 hours in 20 μl 1× CutSmart buffer (NEB), 0.2 μl was quantified using PicoGreen DNA (Life Technologies), and equivalent amounts of DNA separated on 25 cm 1% 1× TBE gels overnight at 90 V. Gels were washed in 0.25 N HCl for 15 minutes, 0.5 N NaOH for 45 minutes, and twice in 1.5 M NaCl, 0.5 M Tris (pH 7.5) for 20 minutes before being transferred to 20 × 20 cm HyBond N + membrane in 6× SSC. Membranes were probed using random primed probes to the rDNA intergenic spacer region in 10 ml UltraHyb (Life Technologies) at 42°C and washed with 0.1× SSC 0.1% SDS at 42°C, or probed in Church Hyb at 65°C and washed with 0.5× SSC 0.1% SDS at 65°C. Images were obtained by exposure to phosphoimaging screens (GE), developed using a FLA 7000 phosphoimager (GE), and quantified using ImageQuant v7.0 (GE). Fig 6A was probed with a biotin-labelled RNA probe to the rDNA intergenic spacer region in 10 ml Ultra-Hyb (Life Technologies) at 42°C and washed with 0.1× SSC 0.1% SDS at 42°C before hybridising with IRDye 680LT Streptavidin (LI-COR) and scanning on an Odyssey DLx (LI-COR). The Pulsed Field Gel in S9 Fig was run on a CHEF DRIII machine (Bio-Rad) using conditions: 0.5× TBE, 1% agarose, 60- to 120-second switch time, 6 V/cm, 14°C, 24 hours run time, then blotted and probed as above using a random primed probe against the *BNA5* gene. Full blot images are presented in S1 Raw Images.

## RNA extraction and RNA-seq library preparation

Cells were resuspended in 50 μl Lysis/Binding Buffer (from mirVANA kit, Life Technologies AM1560), and 50 μl 0.5 μm zirconium beads (Thistle Scientific 11079105Z) added. Cells were lysed with 5 cycles of 30-second 6,500 ms$^{-1}$/30-second ice in an MP Fastprep bead beater or for 3 minutes at power 12 in a Bullet Blender (Thermo Fisher) in cold room, then 250 μl Lysis/Binding buffer added followed by 15 μl miRNA Homogenate Additive, and cells were briefly vortexed before incubating for 10 minutes on ice. 300 μl acid phenol:chloroform was added, vortexed, and centrifuged 5 minutes at 13,000*g*, room temperature before extraction of the upper phase. 400 μl room temperature ethanol and 2 μl glycogen (Sigma G1767) were added and mixture incubated for 1 hour at −30°C before centrifugation for 15 minutes at 20,000*g*, 4°C. The pellet was washed with cold 70% ethanol and resuspended in 10 μl water. 1 μl RNA was glyoxylated and analysed on a BPTE mini-gel, and RNA was quantified using a PicoGreen RNA kit (Life Technologies R11490) or Qubit RNA HS Assay Kit.

150 ng RNA was used to prepare libraries using the NEBNext Ultra II Directional mRNA-seq kit with poly(A)+ purification module (NEB E7760, E7490) as described with modifications: Reagent volumes for elution from poly(T) beads, reverse transcription, second strand synthesis, tailing, and adaptor ligation were reduced by 50%; libraries were amplified for 13 cycles using 2 μl each primer per 50 μl reaction before 2 rounds of AMPure bead purification

at 0.9× and elution in 11 μl 0.1× TE prior to quality control using a Bioanalyzer HS DNA ChIP (Agilent) and quantification using a KAPA Library Quantification Kit (Roche).

## DNA extraction and library preparation for genome resequencing

Cells were resuspended in 60 μl PFGE wash buffer (10 mM Tris HCl (pH 7.5), 50 mM EDTA) with 1 μl lyticase (17 U/ μl in 10 mM KPO$_4$ (pH 7), 50% glycerol, Merck >2,000 U/mg L2524), heated to 50˚C for 2 minutes before addition of 40 μl molten CleanCut agarose (Bio-Rad 1703594), vortexing vigorously for 5 seconds before pipetting in plug mould (Bio-Rad 1703713), and solidifying 15 to 30 minutes at 4˚C. Each plug was transferred to a 2-ml tube containing 500 μl PFGE wash buffer with 10 μl 17 U/μl lyticase and incubated 3 hours at 37˚C. Solution was replaced with 500 μl PK buffer (100 mM EDTA (pH 8), 0.2% sodium deoxycholate, 1% sodium N-lauroyl sarcosine, 1 mg/ml Proteinase K) and incubated overnight at 50˚C. Plugs were washed 4 times with 1 ml TE for 1 to 2 hours at room temperature with rocking; 10 mM PMSF was added to the second and third washes from 100 mM stock (Merck 93482). For 15 minutes, ½ plugs were equilibrated with 1 ml agarase buffer (10 mM Bis-Tris–HCl, 1 mM EDTA (pH 6.5)) and then the supernatant removed and 50 μl agarase buffer added. Plugs were melted for 20 minutes at 65˚C, transferred for 5 minutes to a heating block preheated to 42˚C, 1 μl β-agarase (NEB M0392S) was added and mixed by flicking without allowing sample to cool, and incubation continued at 42˚C for 1 hour. DNA was ethanol precipitated with 25 μl 10 M NH$_4$OAc, 1 μl GlycoBlue, and 330 μl of ethanol and resuspended in 130 μl 0.1× TE. DNA was fragmented in an AFA microTUBE (Covaris 520045) in a Covaris E220 duty factor 10, PIP 175, Cycles 200, Temp 11˚C and then ethanol precipitated. 10 ng DNA was used to prepare libraries using the NEBNext Ultra II DNA kit (NEB E7645S) as described with modifications: Reagent volumes for tailing and adaptor ligation were reduced by 75%; libraries were amplified for 9 cycles using 2 μl each primer per 50 μl reaction before 2 rounds of AMPure bead purification at 0.9× and elution in 11 μl 0.1× TE prior to quality control using a Bioanalyzer HS DNA ChIP (Agilent) and quantification using a KAPA Library Quantification Kit (Roche).

## Sequencing

DNA- and RNA-seq libraries were sequenced by the Babraham Institute Sequencing Facility using a NextSeq 500 instrument on 75 bp single end mode. All RNA-seq data have been deposited at GEO under accession numbers GSE207429 and GSE207503.

## RNA-seq QC, mapping, quantification, and analysis

FASTQ format files provided by the Babraham Institute Sequencing Facility were processed to integer read counts per gene using a Nextflow (v20.01.0) pipeline slightly modified from the RNA-seq pipeline provided by nf-core (v1.4.2) [92,93]. Reads were adapter and quality trimmed using Trim Galore! (v0.6.5) (www.bioinformatics.babraham.ac.uk/projects/trim_galore) wrapper around Cutadapt (v2.10) [94]. Libraries were then assessed for overall quality using FastQC (v0.11.9) (https://www.bioinformatics.babraham.ac.uk/projects/fastqc/). Reads were mapped to the R64-1-1 *S. cerevisiae* genome assembly using HISAT2 (v2.2.0) [95,96]. Mapped reads were quantified over exons and grouped at the gene_id level using feature-Counts (v2.0.0) [97]. For mapping and quantification, we used ENSEMBL release 99 annotation augmented with 4 additional loci between the canonical RNA Polymerase I–transcribed rRNA genes. The following were annotated with the "rDNA_intergenic" gene biotype: IGS2-1S at XII: 485433–459675,—strand; IGS2-1AS at XII: 458433–459675, + strand; IGS1-2S at XII: 459797–460711,—strand; IGS1-2AS at XII: 459797–460711, + strand. For downstream

analyses, we included all genes with annotated gene_biotype: "protein_coding," "transposable_element," or "rDNA_intergenic." We excluded all mitochondrially encoded genes and those around the rDNA locus (i.e., with start or end positions within XII: 450000–491000) apart from ones with the "rDNA_intergenic" biotype. We performed most RNA-seq data manipulations downstream of featureCounts in the R (v4.2.1) programming environment [98]. We made significant use of the Tidyverse (v1.3.1) family of packages [99], particularly stringr (v1.4.0), dplyr (v1.0.9), purrr (v0.3.4), tidyr (v1.2.0), readr (2.1.2), and tibble (v3.1.7). The DESeq2 (v1.36.0) [55] package was obtained from the Bioconductor repository using BiocManager (v3.15) [100].

## RNA-seq normalisation, differential expression testing, and graphical output

In general, we used the built-in DESeq2 size factor normalisation process to make values comparable across sequenced libraries for visualisation as well as for differential expression testing. We applied a BH-corrected FDR cutoff of 0.05 for all differential expression tests. We categorised genes with $\log_2$ fold-change $> 0.5$ as significantly up-regulated and genes with $\log_2$ fold-change $< -0.5$ as significantly down-regulated. For display, we $\log_2$ transformed normalised read counts using an added pseudocount of 1 to deal with any instances of 0 detected counts.

The lists of significantly differentially expressed genes for S1C and S2A Figs were derived from a fitted DESeq2 model with Age (categorical, levels: log phase, 48 hours) as a single design factor. This included log phase and 48-hour-aged samples in triplicate from the wild-type background. Pairwise comparisons mentioned in section "ERC accumulation exerts minimal effects on individual genes" were performed using fitted DESeq2 models comparing 48-hour-aged samples of wild type (triplicate) with *spt3Δ* (triplicate), *rad52Δ* (duplicate), or *sac3Δ* (triplicate) using Genotype (categorical, levels: wild type, mutant) as the single design factor.

Displayed data for Fig 2C as well as the list of genes displayed in heatmap 2B was derived from a fitted DESeq2 model with Age (categorical, levels: log phase, 24 hours, 48 hours), Genotype (categorical, levels: wild type, *spt3Δ*, *rad52Δ*, and *sac3Δ*), ERC abundance (categorical, levels: low, high), and *TRP1* (categorical, levels: absent, present). This included log phase, 24-hour, and 48-hour-aged samples in triplicate for wild type, *TRP1*-repaired wild type, *spt3Δ*, and *sac3Δ* as well as in duplicate for *rad52Δ*. For the ERC abundance factor, most samples were classified as "low," while wild type (including *TRP1*-repaired wild type) 24-hour and 48-hour aged samples were "high." For the *TRP1* factor, wild-type cells had the marker "absent," while all others (including *TRP1*-repaired wild types) had the marker "present." The full set of linear model $\log_2$-fold change coefficients and DESeq2 results are in S1 File.

For analysis of SIR complex-regulated genes, we considered genes to be regulated by the SIR complex if they were determined to be differentially expressed (under the original authors definition) in *sir2Δ*, *sir3Δ*, and *sir4Δ* backgrounds by Ellahi and colleagues (bolded entries in Table 7 from that publication). There were 40 of 42 genes present in our filtered RNA-seq dataset.

Displayed data for Figs 4B, 4C, 4D, S4A, S4B, S4C and S4D as well as the list of genes displayed in Figs 4A, S4A, S6E, and S8C was derived from a fitted DESeq2 model with Age (categorical, levels: log phase, 24 hours, 48 hours), Genotype (categorical, levels: wild type, *spt3Δ*, *rad52Δ*, and *sac3Δ*), *TRP1* (categorical, levels: present, absent), and SEP status (continuous, values min-max scaled to the range [0,1]). Data included as for the previous linear model. *TRP1* categorised as above. Values for SEP status taken from TOM70-GFP medians shown in Figs 3A and S3C with wild type values also used for *TRP1*-repaired samples. The full set of

linear model $\log_2$-fold change coefficients and DESeq2 results are in S2 File, while equivalent information for the modified version fitted with the *rad52Δ* dataset is in S5 File.

Displayed data for Fig 4E and 4F were derived from a fitted DESeq2 model with Age (categorical, levels: log phase, 24 hours, 48 hours), Genotype (categorical, levels: wild type, *spt3Δ*, *rad52Δ*, and *sac3Δ*), TRP1 (categorical, levels: present, absent), SEP status (continuous, values min-max scaled to the range [0,1]), and Carbon source (categorical, levels: glucose, galactose). All previously used data were included as described above and with the "glucose" Carbon source. Additionally, we included 5 sets of log phase, 24-hour, and 48-hour samples of YDH18 cells grown on glucose and 6 sets of the same grown on galactose (data originally from [22]). These were considered to be of the wild-type genotype for modelling due to the negligible genotypic and phenotypic differences. They also had the *TRP1* marker "present." SEP values for used for wild types in the above model were used for glucose-grown DH18 samples. SEP values for galactose-grown DH18 samples were taken from Fig 1B in [22]. The full set of linear model $\log_2$-fold change coefficients and DESeq2 results are in S3 File, while equivalent information for the modified version fitted with the *rad52Δ* dataset is in S6 File.

Displayed RNA-seq data for Figs 1C, 2B, 2D, 4A, S4A and S5E were adapter and quality trimmed using Trim Galore (v0.6.6) and were mapped to yeast genome R64-1-1 using HISAT2 v2.1.0 [96] by the Babraham Institute Bioinformatics Facility. Mapped data were imported into SeqMonk v1.47.0 (https://www.bioinformatics.babraham.ac.uk/projects/seqmonk/) and quantified for $\log_2$ total reads mapping to the antisense strand of annotated open reading frames (opposite strand-specific libraries) with probes included to each strand of the rDNA intergenic spacer regions. Read counts were adjusted by size factor normalisation for the full set of quantified probes.

Hierarchical clustering plots and PCAs were calculated within SeqMonk. MA plots were generated in GraphPad Prism (v9.2.0) comparing mean and difference for each gene between 2 conditions. Probes with less than 2 reads post-normalisation in the control condition were filtered as change between datasets cannot be accurately quantified in this case.

## Gene Ontology term enrichment analysis

GO analysis of individual clusters performed using GOrilla (http://cbl-gorilla.cs.technion.ac.il/) [101]. Quoted *p*-values for GO analysis are FDR-corrected according to the Benjamini and Hochberg method (q-values from the GOrilla output). For brevity, only the order of magnitude rather than the full q-value is given [102]. Full GO analyses are provided in S7 File.

## Genome resequencing mapping and analysis

After adapter and quality trimming using Trim Galore (v0.6.6), DNA-seq data were mapped to yeast genome R64-1-1 using Bowtie 2 [103] by the Babraham Institute Bioinformatics Facility. Reads were quantified in 25 kb windows spaced every 5 kb. Windows overlapping Ty elements and LTRs, mitochondrial DNA, ENA locus, and CUP1 locus were excluded, and then size factor normalisation applied [104]. For rDNA analysis, reads were quantified in 1 kb windows, 200 bp spacing, and size factor normalised.

## Flow cytometry

Cell pellets were resuspended in 240 μl PBS and 9 μl 10% Triton X-100 containing 0.3 μl of 1 mg/ml Streptavidin conjugated with Alexa Fluor 647 (Life Technologies) and 0.6 μl of 1 mg/ml WGA conjugated with CF405S (Biotium). Cells were stained for 10 minutes at room temperature on a rotating mixer while covered with aluminium foil, washed once with 300 μl PBS containing 0.01% Triton X-100, resuspended in 30 μl PBS, and immediately subject to flow

cytometry analysis. Flow cytometry analysis was conducted using an Amnis ImageStream X Mk II with the following laser power settings: 405 = 25 mW; 488 = 180 mW; 561 = 185 mW; 642 = 5 mW; SSC = 0.3 mW.

Cell populations were gated for single cells based on area and aspect ratio ($>$0.8) values, and in-focus cells were gated based on a gradient RMS value ($>$50). Further gating of streptavidin-positive (AF647) cells was also applied, all in a hierarchical manner and 1,000 events acquired. For budding cell analysis, 10,000 events were acquired, and based on focused images and area and aspect ratio, cells were gated into 2 groups: Single cells were gated and counted as G1 phase cells, and Budding cells (doublets or aggregates of 3 cells) were gated and counted as G2/S phase cells. Before data analysis, compensation was applied according to single-colour controls and an automatically calculated compensation matrix. Total fluorescence intensity values of different parameters were extracted using the Intensity feature of the IDEAS software, with Adaptive Erode modified mask coverage. In the analysis, only positive values of fluorescence were included (i.e., where cells were truly positive for the marker) and median values of populations were determined with Graphpad Prism (v9.2.0).

## Statistical analysis

All statistical analysis other than that performed using the R DESeq2 package was performed in GraphPad Prism (v9.2.0).

## Supporting information

**S1 Fig. Supplement to ERCs do not cause genome-wide gene expression dysregulation.** (**A**) Southern blot analysis of ERC abundance in log phase and 48-hour-aged wild-type and *fob1Δ* MEP cells. Quantification shows ratio between sum of ERC bands and chromosome XII band for 3 biological replicates, *p*-value calculated by 1-way ANOVA. Images show 2 sections of the same blot membrane with no differential image processing applied. (**B**) Southern blot analysis of ERC abundance at log phase and indicated aged time points for wild type, *rad52Δ*, *spt3Δ*, and *sac3Δ* mutant MEP cells, along with a wild type in which the *TRP1* gene has been restored. (**C**) Scatter plot of $\log_2$ mRNA abundance comparing log phase to 48-hour-aged wild type. Sets of genes called as significantly differentially expressed by DESeq2 (BH-corrected FDR $<$ 0.05, $\log_2$-fold change threshold ±0.5) are highlighted in colour. GO terms are provided for each category, significance indicated by FDR-corrected q values, full GO term enrichment analysis in S7 File. (**C**) PCA summary for log phase poly(A)+ RNA-seq libraries from wild type and *rad52Δ*, *spt3Δ*, and *sac3Δ* mutants. (**E**) Hierarchical clustering analysis of the 2,606 genes significantly differentially expressed at log phase between at least 1 mutant and the wild type by DESeq2 (BH-corrected FDR $<$ 0.01, $\log_2$-fold change threshold ±0.5). Individual biological replicates are shown; note that scale is different to other hierarchical clustering analyses presented as differences between log phase samples are smaller than age-linked changes. The numerical data underlying this Figure can be found in S8 File. BH, Benjamini–Hochberg; ERC, extrachromosomal ribosomal DNA circle; FDR, false discovery rate; GO, gene ontology; MEP, mother enrichment program; PCA, principal component analysis. (TIF)

**S2 Fig. Supplement to expression change of individual genes is not associated with ERC accumulation.** (**A**) Schematic of known RNA polymerase II transcripts in the intergenic spacer region of the rDNA, and quantification of noncoding transcripts orientated sense and antisense to the RNA polymerase I–transcribed 35S genes. Individual biological replicates are shown as points (3 for wild type, *spt3Δ*, and *sac3Δ* and 2 for *rad52Δ*), and bars show mean

values for a given age and genotype. (**B**) Distribution of $\log_2$ mean normalised mRNA abundance for all genes in wild-type cells, including probes to rDNA intergenic spacer regions. Red arrows highlight the probe that detects the IGS1-F noncoding RNA as defined in panel A. (**C**) Coefficients representing the contribution of each individual DESeq2 linear model factor to mean $\log_2$ normalised mRNA abundance of cluster 2 genes (from Fig 2B) across all datasets. DESeq2 was used to model expression as dependent on 4 categorical independent variables (with an additional intercept factor): Genotype (wild type, *spt3Δ*, *rad52Δ*, *sac3Δ*), *TRP1* auxotrophy (present, not present), Age (log phase, 24 hours, 48 hours), ERC accumulation (low, high). Coefficient values can be interpreted as the $\log_2$ normalised mRNA abundance change caused by a specific model factor being present relative to a wild-type log phase *TRP1* auxotroph with low ERC levels. Solid horizontal lines within violins indicate median values, while lower and upper quartiles are indicated by dotted horizontal lines. Full output of the model is provided in S1 File. (**D**) As for C showing values for the 40 SIR-complex regulated genes (from Ellahi and colleagues) across all datasets. The numerical data underlying this Figure can be found in S8 File. ERC, extrachromosomal ribosomal DNA circle; rDNA, ribosomal DNA. (TIF)

**S3 Fig. Supplement to mutants lacking ERCs still display signatures of the SEP.** (**A**) Comparison between manual bud scar counts and median WGA intensities for wild type, *spt3Δ*, and *rad52Δ* at 24- and 48-hour time points. $n = 15$ cells for bud scars; $n = 4$ medians of biological replicate populations for WGA; error bars ±1 SD. (**B**) Analysis of Rpl13a-mCherry to show ribosomal protein abundance. Samples and analysis as in Fig 3A. (**C**) Plots of Tom70-GFP, Rpl13a-mCherry, WGA, and cell size for wild-type and *sac3Δ* cells at indicated ages, obtained as in Fig 3A. (**D**) Colony size assays measuring fitness of wild type and *spt3Δ* cells aged in glucose. Log phase cells and cells aged 48 hours then purified live in media were placed on YPD plates by micromanipulation. Colony diameters were measured on the micromanipulator screen after 24 hours. Only cells that formed visible colonies after a further 3 days were included in the analysis. Log phase cells were taken directly from culture and colony growth measured under the same conditions. *p*-Values calculated by Kruskal–Wallis test, $n = 18$ for log wild type, 25 for log *spt3Δ*, 40 for aged wild type, 22 for aged *spt3Δ*. (**E**) Distribution of signal areas from flow cytometry images of Rpa190-GFP in populations of log phase cells or mother cells aged for 24 and 48 hours. A total of 1,000 cells were imaged per population after gating for circularity (all) and biotin (aged cells only, based on streptavidin-Alexa647); images were post-filtered for focus of Rpa190-GFP. (**F**) Distribution of signal intensities from flow cytometry images of Vph1-mCherry in populations of log phase cells or mother cells aged for 24 and 48 hours. A total of 1,000 cells were imaged per population after gating for circularity (all) and biotin (aged cells only, based on streptavidin-Alexa647); images were post-filtered for focus of Vph1-GFP. (**G**) Example images of unseparated clusters of cells (brightfield) originating from single aged mother cells (streptavidin stained, red). The numerical data underlying this Figure can be found in S8 File. ERC, extrachromosomal ribosomal DNA circle; SEP, senescence entry point; WGA, wheat germ agglutinin. (TIF)

**S4 Fig. Supplement to a gene expression signature for the SEP.** (**A**) Hierarchical clustering of $\log_2$ mRNA abundance for 290 genes called by DESeq2 Linear Model 2 as significantly different between datasets based on age. DESeq2 modelling as described for 4B, clustering procedure as for 2B. Individual biological replicates shown (3 for wild type, *spt3Δ*, and *sac3Δ* and 2 for *rad52Δ*). (**B**) Coefficients representing the contribution of each individual parameter to the differences in mRNA abundance of the set of genes significantly differentially expressed with Tom70-GFP levels across all datasets calculated using DEseq2 linear model 2 as described for

4B. (**C**) Log$_2$ change of mRNA abundance from log phase to given time points for all genes on ChrXIIr (left) or all genes on chromosomes other than XII (right). Individual biological replicates are shown; boxes show median and interquartile range; whiskers show upper and lower deciles. (**D**) Medians of mRNA abundance change datasets (Fig 4C) for ChrXIIr and other chromosomes. Wild type and TRP1 datasets are pooled to give $n = 6$ per condition, $p$-values calculated by 1-way ANOVA with post hoc Tukey test. Red $p$-values indicate significance at $p < 0.05$. (**E**) Distribution of signal intensities from flow cytometry images of Tom70-GFP and WGA-Alexa405 in populations of haploid MATα log phase cells or mother cells aged for 24 and 48 hours in YPD. A total of 1,000 cells are imaged per population after gating for circularity (all) and biotin (aged cells only, based on streptavidin-Alexa647). Images are post-filtered for focus of Tom70-GFP, leaving approximately 600 cells per population. (**F**) Log$_2$ change of mRNA abundance for haploid MATα MEP cells from log phase to given time points for all genes on ChrXIIr (left) or all genes on chromosomes other than XII (right). (**G**) As Fig 4B excluding genes on ChrXIIr. The numerical data underlying this Figure can be found in S8 File. MEP, mother enrichment program; SEP, senescence entry point.
(TIF)

**S5 Fig. Connecting SEP-associated gene expression change with stress and growth.** (**A**) Correlations between coefficients for SEP and Age, representing the contribution of these variables to gene expression change during ageing calculated by Linear Model 3, and gene expression change during a 20-minute heat shock (GSE18). Only genes participating in the ESR are included [13]. $R^2$ values calculated by linear regression. (**B**) Correlation between coefficients for the SEP, representing the contribution of this variable to gene expression change during ageing calculated by Linear Model 3, and gene expression difference between specific growth rates μ = 0.1 and μ = 0.33 [66]. Only genes participating in the ESR are included [13]. $R^2$ value calculated by linear regression; n is number of genes analysed as not all ESR genes were measured by Regenberg and colleagues. (**C**) Summary of correlations between coefficients for SEP and Age, representing the contribution of these variables to gene expression change during ageing calculated by Linear Model 3, and gene expression change during different stresses (GSE18). $R^2$ values were calculated for the comparison of each individual stress dataset reported by Gasch and colleagues and the SEP/Age coefficients. $R^2$ values were then binned into 8 different stress types, and $p$-values representing the differences between correlations to the SEP and Age coefficients calculated by 1-way ANOVA. (**D**) Correlation between gene expression change in ageing (wild type log phase to 48 hours) and in heat shock (GSE18) for the set of 187 genes significantly associated with the SEP by Linear Model 2. $R^2$ value calculated by linear regression; n is number of genes analysed as not all the 187 SEP genes were measured by Gasch and colleagues. (**E**) Correlation between gene expression change in ageing (wild-type log phase to 48 hours) and between specific growth rates μ = 0.1 and μ = 0.33 [66] for the set of 187 genes significantly associated with the SEP by Linear Model 2. $R^2$ value calculated by linear regression; n is number of genes analysed as not all the 187 SEP genes were measured by Regenberg and colleagues. The numerical data underlying this Figure can be found in S8 File. ESR, environmental stress response; SEP, senescence entry point.
(TIF)

**S6 Fig. Aneuploidies emerge during ageing.** Genome resequencing data and RNA-seq data as in Figs 5C and 4C, respectively, showing mild accumulations of Chromosomes I and VI during ageing. The numerical data underlying this Figure can be found in S8 File.
(TIF)

**S7 Fig. Ageing of *hst3Δ hst4Δ* is equivalent to *rad52Δ*.** (**A**) Cell cycle stage plots comparing the percentage of budded and unbudded cells. Imaging flow cytometry was performed with gating for streptavidin (aged samples only), then budded and unbudded cells called based on circularity and validated by examining representative images. (**B**) Imaging flow cytometry analysis of Tom70-GFP intensity, which increases hugely in G1 cells after the SEP [5], WGA, which reports on replicative age, and cell size in wild type and *hst3Δ hst4Δ*. Cells were gated for streptavidin staining (aged samples only), which filters out young cells, then for circularity, which selects towards G1 cells and removes clumps. Graph shows medians of biological replicate populations; *p*-values calculated from 2-way ANOVA with post hoc Tukey test using genotype and age as independent variables, *n* = 2. Red *p*-values indicate significance at $p < 0.05$. (**C**) Southern blot analysis of ERC abundance in log phase and 24-hour-aged wild-type and *hst3Δ hst4Δ* MEP cells; quantification shows ration of ERC bands to chromosomal rDNA. Individual lanes were spliced from the same blot image; no differential image processing was applied. (**D**) MA plots comparing $\log_2$ mRNA abundance distributions between log phase and 48-hour-aged samples from wild type and indicated mutants. x-Axis is $\log_2$ average normalised read count; y-axis is change in $\log_2$ normalised read count from young to old. Slope is calculated by linear regression. Different matched wild-type control data are presented for each mutant; *rad52Δ* data are provided as a comparison for *hst3Δ hst4Δ*. Data for each genotype and age calculated as the $\log_2$ mean normalised read counts per gene of 2 biological replicates for mutants, 3 biological replicates for wild type. (**E**) Hierarchical clustering of $\log_2$ mRNA abundance for 187 genes called by a DESeq2 Linear Model 2 as significantly different between datasets based on the SEP, as in Fig 4A, for wild type, *rad52Δ*, and *hst3Δ hst4Δ* at log and 24 hours, 2 biological replicates per time point for mutants, 3 biological replicates for wild type. (**F**) $\log_2$ change of mRNA abundance from log phase to given time points for all genes on ChrXIIr (left) or all genes on chromosomes other than XII (right). Data are the mean of 2 biological replicates. (**G**) MA plots comparing $\log_2$ mRNA abundance distributions between log phase and 48-hour-aged samples from wild type and *mus81Δ*. x-Axis is $\log_2$ average normalised read count; y-axis is change in $\log_2$ normalised read count from young to old. Slope is calculated by linear regression. Matched wild-type control data are presented, data for each genotype and age calculated as the $\log_2$ mean normalised read counts per gene of 2 biological replicates. The numerical data underlying this Figure can be found in S8 File. ERC, extrachromosomal ribosomal DNA circle; MEP, mother enrichment program; SEP, senescence entry point; WGA, wheat germ agglutinin.
(TIF)

**S8 Fig. Linear model outcomes are minimally influenced by exclusion of *rad52Δ* dataset.** (**A**) Fraction of genes significantly differentially expressed based on SEP variable discovered by Linear Models 2 and 3 fitted using either the full dataset of wild type, *spt3Δ*, *sac3Δ*, and *rad52Δ*, or a reduced dataset of just wild type, *spt3Δ*, and *sac3Δ*. Values are observed number of genes per chromosome divided by the expected number of genes (calculated from the total number of significantly differentially expressed genes × genes on chromosome / genes in genome). Chromosome XII values are split between XII left–the region from the left telomere to the rDNA, and XIIr–from the rDNA to the right telomere (ChrXIIr). Full output of the models is provided in S4 File and S5 File. (**B**) Assignment of gene expression change in wild-type ageing log to 48 hours to SEP and Age contributions calculated by Linear Model 3 fitted without *rad52Δ* dataset, displayed as MA plots. Analysis as in Fig 4F; left panel shows wild-type ageing data for comparison and is identical to the left-hand plot in Fig 4F. The numerical data underlying this Figure can be found in S8 File.
(TIF)

**S9 Fig. Supplement to increased copy number of genes telomere proximal to the rDNA on ChrXII is not responsible for the SEP.** (**A**) Pulsed field gel electrophoresis analysis showing successful translocation of ChrXIIr onto chromosome V. (**B**) MA plots comparing $\log_2$ mRNA abundance distributions between log phase and 48-hour-aged samples from **a:a** diploid wild-type and Chr XII <>V translocation strain. x-Axis is $\log_2$ average normalised read count; y-axis is change in $\log_2$ normalised read count from young to old. Slope is calculated by linear regression. (**C**) Hierarchical clustering of $\log_2$ mRNA abundance for 187 genes called by a DESeq2 Linear Model 2 as significantly different between datasets based on the SEP, as in Fig 4A, excluding genes on ChrXIIr, for **a**: **a** diploid wild-type and Chr XII <>V translocation strain. The numerical data underlying this Figure can be found in S8 File. rDNA, ribosomal DNA; SEP, senescence entry point.
(TIF)

**S10 Fig. Model for the formation mechanism of ChrXIIr.** (**A**) Replication fork dynamics leading to incomplete replication at anaphase. (i) Potential replication fork stalling elements in the rDNA include the RFB, DNA damage, and regions highly expressed by RNA polymerase I, II, and III. (ii) First replication fork stalls at the RFB or on collision with RNA polymerase I. (iii) Converging replication fork stalls due to DNA damage, collision with transcription units, topological strain, etc. (iv) Centromeres are pulled apart on the mitotic spindle, but segregation cannot be completed. (v) Cleavage of replication fork structures by structure-specific endonucleases allows segregation to complete but forms ChrXIIr. (**B**) ChrXIIr formation in the context of heterozygous markers. Incomplete replication leads to one chromosome forming a bridging chromosome, resolution of which either leaves both chromosome fragments in one nucleus (left), in which case the break can be repaired, or separates the fragments (right) in which case ChrXIIr formation in the mother is accompanied by loss of heterozygosity in the daughter. Further recombination events between ChrXIIr and full-length chromosome XII in the mother can lead to loss of heterozygosity also in the mother. rDNA, ribosomal DNA; RFB, replication fork barrier.
(TIF)

**S1 Table. Strains used in this work.** All strains are in the MEP background; most are diploid as indicated. Wild-type alleles are indicated as +; fluorophore-tagged constructs are heterozygous to avoid growth defects.
(DOCX)

**S2 Table. Oligonucleotides used in this work.**
(DOCX)

**S3 Table. Plasmids used in this work.**
(DOCX)

**S1 File. Details and output data for Linear Model 1.** Description of the variables used, calculated Model Coefficients and Significance tables for each comparison.
(XLSX)

**S2 File. Details and output data for Linear Model 2.** Description of the variables used, calculated Model Coefficients and Significance tables for each comparison.
(XLSX)

**S3 File. Details and output data for Linear Model 3.** Description of the variables used, calculated Model Coefficients and Significance tables for each comparison.
(XLSX)

**S4 File. Genome resequencing data for individual chromosomes.** Change in age for individual chromosomes in each mutant, data processed as in Fig 5C. Scales have been kept the same as in Fig 5C to allow easy comparison.
(PDF)

**S5 File. Details and output data for Linear Model 2 fitted without *rad52Δ*.** Description of the variables used, calculated Model Coefficients and Significance tables for each comparison.
(XLSX)

**S6 File. Details and output data for Linear Model 3 fitted without *rad52Δ*.** Description of the variables used, calculated Model Coefficients and Significance tables for each comparison.
(XLSX)

**S7 File. GO analysis for differentially expressed gene sets.**
(XLSX)

**S8 File. Numerical data underlying Figures.**
(ZIP)

**S1 Raw Images. Full unmodified gel images underlying figure panels.**
(PDF)

## Acknowledgments

We would like to thank Rachael Walker and Attila Bebes of the Babraham Institute Flow Facility for their assistance with imaging flow cytometry; Paula Kokko-Gonzales and Nicole Forrester of the Babraham Institute Next Generation Sequencing facility for sequencing samples and QC assistance; and Felix Krueger, Laura Biggins, and Anne Segonds-Pichon of the Babraham Institute Bioinformatics Facility for help with sample processing and statistical analysis. We thank Michelle King for Southern blot analysis; Patrick Lindstrom and Dan Gottschling for the MEP system; Gilles Charvin and Théo Aspert for helpful discussions; Luca Love and Prasanna Channathodiyil for RNA-seq data; and Nianshu Zhang and Nazif Alic for helpful suggestions regarding DESeq2 linear modelling.

## Author Contributions

**Conceptualization:** Andre Zylstra, Jonathan Houseley.

**Formal analysis:** Andre Zylstra.

**Funding acquisition:** Jonathan Houseley.

**Investigation:** Andre Zylstra, Hanane Hadj-Moussa, Dorottya Horkai, Alex J. Whale, Baptiste Piguet, Jonathan Houseley.

**Methodology:** Alex J. Whale.

**Software:** Andre Zylstra.

**Visualization:** Andre Zylstra.

**Writing – original draft:** Andre Zylstra, Jonathan Houseley.

**Writing – review & editing:** Andre Zylstra, Hanane Hadj-Moussa, Dorottya Horkai, Baptiste Piguet, Jonathan Houseley.

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
