## [Editor Report · Decision Letter 0]

26 Aug 2022

Dear Jon, 

Thank you for submitting your manuscript entitled "Senescence in yeast is associated with chromosome XII cleavage rather than ribosomal DNA circle accumulation" for consideration as a Research Article by PLOS Biology.

Your manuscript has now been evaluated by the PLOS Biology editorial staff, as well as by an academic editor with relevant expertise, and I'm writing to let you know that we would like to send your submission out for external peer review.

Once your full submission is complete, your paper will undergo a series of checks in preparation for peer review. After your manuscript has passed the checks it will be sent out for review. To provide the metadata for your submission, please Login to Editorial Manager (https://www.editorialmanager.com/pbiology) within two working days, i.e. by Aug 30 2022 11:59PM.

Kind regards,

Roli

Roland Roberts, PhD

Senior Editor

PLOS Biology

rroberts@plos.org

---

## [Decision Letter · Decision Letter 1]

31 Oct 2022

Dear Jon,

Thank you for your patience while your manuscript "Senescence in yeast is associated with chromosome XII cleavage rather than ribosomal DNA circle accumulation" was peer-reviewed at PLOS Biology. Your manuscript has been evaluated by the PLOS Biology editors, an Academic Editor with relevant expertise, and by four independent reviewers.

As with the other manuscript, there's some significant positivity here, but again some substantial concerns which will need to be addressed. You'll see that reviewer #1 is concerned about causality here, and requests an experiment (the other issues are textual). Reviewer #2 finds this paper frustrating, as it doesn’t deliver a new model; s/he suggests a significant number of experiments but is not optimistic that the problems can be addressed satisfactorily. Reviewer #3 raises the same main concern as for the other paper (the reliance on transcriptomics), and worries about the use of rad52∆ as a model for aging and of Tom70-GFP as a marker for SEP. Reviewer #4 thinks that the findings are exciting if true, but again shares several of reviewer #3's concerns, and has some significant experimental requests. In case it's helpful, I can reveal that the reviewers appear in the same order in the two decision letters.

As you will see in the reviewer reports, which can be found at the end of this email, although the reviewers find the work potentially interesting, they have also raised a substantial number of important concerns. Based on their specific comments and following discussion with the Academic Editor, it is clear that a substantial amount of work would be required to meet the criteria for publication in PLOS Biology. However, given our and the reviewer interest in your study, we would be open to inviting a comprehensive revision of the study that thoroughly addresses all the reviewers' comments. Given the extent of revision that would be needed, we cannot make a decision about publication until we have seen the revised manuscript and your response to the reviewers' comments. Your revised manuscript would need to be seen by the reviewers again, but please note that we would not engage them unless their main concerns have been addressed. 

We appreciate that these requests represent a great deal of extra work, and we are willing to relax our standard revision time to allow you 6 months to revise your study. Please email us (plosbiology@plos.org) if you have any questions or concerns, or envision needing a (short) extension.

**IMPORTANT - SUBMITTING YOUR REVISION**

*Resubmission Checklist*

*Published Peer Review*

*PLOS Data Policy*

*Blot and Gel Data Policy*

Sincerely,

Roli

Roland Roberts, PhD

Senior Editor

PLOS Biology

rroberts@plos.org

REVIEWS' COMMENTS:

Reviewer #1:

Numerous studies in budding yeast have demonstrated a strong correlation between aging and the accumulation of extra-chromosomal ribosomal circles (ERCs). Indeed, high levels of ERCs are widely believed to be the proximal driver of senescence, although a causal link remains unsubstantiated. In the current manuscript, the authors use mutants that are incapable of either generating or accumulating ERCs to explore the issue of causality. They show convincingly that ERC accumulation is separable from transcriptional and microscopic hallmarks of replicative aging. Furthermore, their transcriptional analyses, combined with previously published work, led them to identify a fragment of the right arm of chrXII (chrXIIr) that can arise due to replication difficulties at the rDNA. The authors show that increased levels of chrXIIr are more tightly associated with hallmarks of aging than are high levels of ERCs. This is not only interesting in itself, but it also makes discoveries about the biology of aging in yeast more relevant to that in higher eukaryotes, which don't accumulate ERCs. 

The work is done carefully, and the topic will be of broad interest to readers of PLoS Biology. My major concern with the manuscript is that it left me unclear about what the authors believe to be the causal relationship between the chrXIIr fragment and replicative senescence. They have clearly demonstrated that hallmarks of aging can occur in the absence of ERC accumulation, thereby proving that ERC accumulation is not required for aging. They have also shown a correlation between appearance of the chrXIIr fragment and aging phenotypes. However, as far as I can tell, just as they argue that correlation does not imply causation with ERCs, the same thing can be argued with the chrXIIr fragment. (The chrXIIr fragment is more tightly associated with the senescence entry program, or SEP, than are ERCs, but the SEP has not been demonstrated to be causal for age-related death.) This in no way nullifies the importance and novelty of the current work, but the authors must explain their model better. 

At different points in the manuscript, the authors contradict themselves about whether chrXIIr is causal for aging. For example, in the introduction, they write: 

"... we implicate a large fragment of chromosome XII (ChrXIIr) as the unanticipated driver of ageing phenotypes in yeast."

while in the discussion they write: 

"... ChrXIIr and ERCs can only be mediators of specific pathology rather than primary drivers of ageing ...."

In addition to clarifying their thoughts about the role of chrXIIr in aging, the authors must perform the experiment described in point 1. Below that, I make suggestions and comments to improve this very interesting manuscript. 

1. Make a MATa/MATa diploid with the XII<>V translocation strain and compare this to a WT (no translocation) MATa/MATa diploid with regard to (a) replicative lifespan; (b) accumulation of chrXIIr during aging (MEP); and (c) appearance of Tom70 foci during aging (MEP). Transcriptomic analyses of these two strains would be nice as well, but not required. 

2. The authors write: "In the XIIr <> V strain, ChrXIIr gene mRNAs did not increase in abundance with age, confirming that this effect arises through rDNA induced chromosomal instability (Figure 5E)." The translocation will also affect rDNA induced chromosomal instability, because when the right portion of chrXII is attached to chrV, it is no longer subject to being cut off of the end of the chromosome due to replication problems at the rDNA. 

3. The authors should include a statement explicitly stating what they believe the proximal cause of aging-related death is in strains that accumulate chrXIIr. Is it loss of heterozygosity (LOH)? Altered composition of the cell wall? Anaphase chromosome bridges due to incomplete replication at the rDNA?

4. The authors should make clear that the senescence entry program (SEP) has not been established as the ultimate cause of aging-associated death. I have the impression that the authors believe that, by showing that chrXIIr is more tightly associated with the SEP than are ERCs, they have shown that chrXIIr is the "real" cause of aging, whereas ERCs are a bit of a red herring that are associated with aging but are not causal. I think that they are correct on the latter point, but it's important to keep in mind that phenotypes like accumulation of Tom70 foci could also be associated with, but not causal to, aging-related death. 

5. Early on, the authors should point out that they are working with diploids, and also add a comment about why they use diploids (e.g. it allows cells to remain viable even following a LOH event). An appropriate place to insert this information would be here: "To investigate the role of ERC accumulation during replicative ageing in S. cerevisiae, we examined aged mutant cells that fail to accumulate ERCs during ageing"

6. As I understand it, there are 187 genes (rows) in the heat map in Figure 4A. It would be helpful to mention this in the legend, since it allows the reader to associate the various numbers of genes mentioned in the text with what is shown in the figures. 

7. Mention whether the following RNA is c-pro or e-pro (or neither), since these are names that some people are familiar with (I believe that it is e-pro): "In fact, the IGS1-F non-coding RNA was induced sufficiently to become the most abundant poly(A)+ RNA in aged wild-type cells (Figure S2B)."

8. Label the gray box on the right side of figure 2C being a key to the symbols in the main plot, since otherwise it looks like a figure in itself. 

Reviewer #2:

The authors explore the question of whether ERCs are causal in aging in budding yeast, which is a long held model in the field. The authors use the strategy of taking three yeast mutants with different levels of ERCs combined with the mother enrichment program (MEP) to see how ERCs contribute to aging. As a read out they rely on several metrics, like bud scars (WGA) to measure number of divisions, accumulation of the mitochondrial protein Tom70-GFP, and gene expression analysis. Their results are not consistent with the model. The data seems very robust with many replicates and statistics. The figures are very nice. They report on a new chromosomal element distal to the rDNA array on chr 12 that accumulates with age. They make a designer chromosome that prevents the occurrence of this species, but SEP still occurs. Their study certainly shoots a lot of holes in the model, but the new chromosomal element doesn't seem to be causal either. Although I tend to agree with the authors, based on the weight of the evidence presented, that the current ERC model is likely incorrect, the end result is somewhat unsatisfying since a new improved model doesn't really emerge, and the chromosomal species isn't demonstrated on a pulsed field gel, which seems essential. They focus the ending model on how the chromosomal species forms rather than an aging model, which is what most of the manuscript drives toward.

Some of the main problems are 1) the pleiotropic effects the deletions will have on gene expression and cell physiology, 2) the heavy reliance on specific markers i.e. Tom70-GFP without explaining why this specific marker should be trusted in all mutant contexts, 3) the complexity of the multiple gene expression models which are difficult to understand, 4) the high demand on the reader to keep track of many correlating phenotypes and for relevant background knowledge, 5) the lack of a new model to explain aging, 5) the lack of evidence for the chromosomal species. The title implies "chromosome XII cleavage" is critical for senescence which doesn't seem appropriate since cleavage itself isn't measured, how the species is generated is only a model, and a strain that doesn't generate the species still shows senescence according to Tom70-GFP.

I have made specific suggestions below for improvement. However, the resolution of these issues may be insufficient to justify publication since some of my key problems relate to the larger issue of the story the authors are trying to tell with the data that they currently have.

Specific suggestions

1. I suggest adding a small annotation to each mutant to indicate ERC hi or ERC low so that reader does not have to keep referring back to figure S1. ERCs in the mutant strains should be quantified as in S1A. In some of the figures it might help to plot the lack of correlation between ERCs and the phenotype being described.

2. If there were other SAGA or TREX mutant have accumulate ERCs to a high level, and their gene expression profiles were similar to spt3 or sac3, this would strengthen the gene expression argument.

3. In figure 2B it makes more sense tome to put the spt3 and sac3 profiles together.

4. The authors use mutations that disrupt gene expression due to effects on RNAPII. How do we know that the Tom70 phenotype cannot be attributed to different effects in each mutant?

5. The heavy reliance on Tom70-GFP as a metric for SEP needs to be justified. In fact in Horkai et al the introduction mentions a trajectory that is not associated with Tom70 accumulation. 

6. The explanation of the gene expression analysis in the last paragraph on page 5 is difficult to understand.

7. How well accepted is gene expression as a metric of aging?

8. The authors use "non-ChrXIIr genes" as a control in several experiments, but it would be important to report also on using the rest of ChrXII as a control, since using the entire genome could easily wipe out non-euploid metrics.

9. The authors refer to ChrXIIr as extrachromosomal, but I don't see evidence to support this claim. The fragment could remain connected to ChrXII.

10. The fragment needs to be visualized on a PFG.

11. As the amplification of ChrXIIr does not occur in the engineered strain (Fig 5) but the SEP still occurs based on Tom70, how can the authors conclude the region is associated with SEP as marked by Tom70-GFP??? What aspects of the expression profile persist with the engineered chromosome? This engineered chromosome is a key reagent to the story.

A few typos:

Figure 2D legend does not mention the cell clusters

Figure 6 labels, "specific" not secific

Figure 2, S2 title, expression change not charge

Reviewer #3:

In the manuscript "Senescence in yeast is associated with chromosome XII cleavage rather than ribosomal DNA circle accumulation" Zylstra et al. challenge the widely accepted view that extrachromosomal rDNA circles (ERCs) are major contributors to cellular senescence and propose instead that accumulation of a fragment of chromosome XII is strongly associated with aging. This conclusion is based on the analysis of purified populations of aged mother cells of different mutant strains that accumulate less or no ERCs. In these populations, the authors perform transcriptome analysis and correlate it with different markers of cell senescence, such as the accumulation of the outer mitochondrial protein Tom70 or cell size.

In my view, the evidence suggesting that ERCs play only a minor role in cell senescence is not conclusive and not sufficient to refute previous evidence showing a strong association between the senescence transition and ERC accumulation, in particular from recent single cell-based studies (PMID: 31291577, PMID: 30042134, PMID: 32675375). The observation that a part of chromosome XII (XIIr) accumulates in old cells is interesting but not per se novel (PMID: 24532716), which the authors acknowledge, but whether and how XIIr accumulation contributes to aging has not been clear. The authors have developed an elegant tool to investigate the functional consequence of XIIr accumulation using chromosome engineering. However, preventing accumulation of XIIr does not alter the investigated aging hallmarks, and is thus rather a consequence than a driving force of aging. This leaves the interesting question of why accumulation of this chromosomal fragment is so tightly associated with aging cells. I believe that especially the role of this recurrent chromosomal abnormality is interesting to the PLOS Biology readership, but I have a number of concerns, some of which I also outlined in detail in the review to the accompanying manuscript "Dietary change without caloric restriction maintains a youthful profile in ageing yeast" from the same group. The first point applies to both manuscripts and parts of it are a direct copy.

1) My main concern is the interpretation of the transcriptome data. The indicated age dependent decrease in ribosome biogenesis genes is reminiscent of activation of the environmental stress response (ESR), which is a hallmark of aging, slow growth or permanent cell cycle arrest (PMID: 22498653, PMID: 17105650, PMID: 30739799). A terminal cell cycle arrest is entered at the end of replicative life irrespective of why and how fast cells aged. The fraction of terminally arrested cells in each population of purified aged mother cells could thus dominate the transcriptome. rad52∆ mutant cells for example have a much shorter RLS than wild type cells (11 vs 25-30, (Delaney et al 2013)). That means that at the 24h timepoint, all rad52∆ cells are terminally arrested while the wild type cells are still all cycling, while at the 48h timepoint probably most cells have entered a terminal cell cycle arrest. If the gene expression profile in aged populations is a consequence of arrested cells, it changes the interpretation of the transcriptome data. For example: If rad52∆, spt3∆ and sac3∆ mutants all display similar age-dependent gene expression changes after 48h of aging as wild type cells, this could mean that by that point they have mostly entered a terminal cell cycle arrest and not that they all age in the same way. Such data can therefore not be used to rule out the contribution of ERCs to aging in wild type cells. The authors need to control their transcriptome data for this cell cycle arrest effect by A) determining division times and the fraction of terminally arrested cells in the aged populations and B) comparing what part of the gene expression change is driven by activation of an ESR, for example by comparing to published data sets of slow growing or arrested cells.

2) Because of the complications described above, a rad52∆ mutant (and other DNA damage repair mutants) is a particularly poor model to study aging because cells stop dividing after roughly 10 divisions and mostly arrest as budded cells (indicative of G2/M arrest potentially linked to persistant DNA damage), in contrast to wild type cells that primarily arrest in G1 after 20-30 divisions (Delaney et al 2013). Despite the problematic use of rad52∆ as aging model, several interpretations strongly depend on the rad52∆ mutant strain, which often shows stronger and opposite phenotypes compared to spt3∆ and sac3∆. For example, accumulation of Tom70-GFP is accelerated in rad52∆ but occurs more slowly in spt3∆ and sac3∆ leading the authors to conclude that ERC accumulation and senescence transition are not related, because all of the above strains have no or reduced levels of ERCs. However, if the rad52∆ data were excluded from this analysis, ERC accumulation and Tom70-GFP accumulation would correlate. Similarly, age dependent changes in gene expression (Figure 2B, 2D, 4A) are often going in opposite directions in rad52∆ than in spt3∆ and sac3∆ mutants. While these data do not necessarily support a role of ERCs in affecting gene expression or the senescence transition, I don't think they present evidence against it either. Because of the above-mentioned issues with the rad52∆ mutant, I am very skeptical about results that are solely supported by the rad52∆ results.

3) Li et al 2020 (PMID: 32675375) have shown that budding yeast cells can follow different trajectories during replicative aging and identified at least two different modes characterized by either accumulation of nucleolar markers (indicative of ERC accumulation) or accumulation of aggregated mitochondria. Cells that loose mitochondrial function tend to slow down cell divisions earlier and stop dividing earlier. These aging trajectories are influenced by genotype, in particular mutant cells that prevent ERC accumulation follow the "mitochondria" aging trajectory. Thus, while Tom70-GFP may be a good marker for SEP in some cells, this may not be the case for all aging cells. To directly measure SEP, the authors should determine cell cycle durations in populations of aged cells. 

4) The observed accumulation of a fragment of Chromosome XII is interesting. In particular, the XIIr <> V chromosome engineering experiment is very elegant and demonstrates that XIIr accumulation does not cause SEP itself. It would be interesting to further investigate the role of this chromosome fragment: If it is acentromeric, as suggested by the authors, do multiple copies accumulate in old mother cells (could be analyzed in single cells using FISH)? If not, is chromosome XII fragmentation a terminal event in replicative aging? An interesting idea would be that one cause of cell death in aged cells are irreparable recombination intermediates in the rDNA. This could be the reason why a fraction of old cells (30% in wt, 70% in rad52∆) arrest as budded cells and would be compatible with the observation that swaping the telomere proximal region of Chromosome XII had no effect on aging. It would predict that the telomere proximal region of Chromosome V is not associated with aging.

Reviewer #4:

In this manuscript, Zylstra et al studied yeast aging in the presence/absence of ERCs, and found that accumulation of ChrXIIr (instead of ERCs) was the primary driver of yeast aging. As their markers, the authors use Tom70-GFP showing the onset of cell division defects at the SEP, and gene expression differences/similarities measured by RNAseq in different genetic backgrounds allowing/blocking formation or accumulation of ERCs.

This is a provocative manuscript with novel claims. If validated by further experiments/controls, this work would mean a paradigm shift in the yeast aging field and would make the yeast model of aging more relevant to aging in higher eukaryotes. Despite all these (potential) excitement, I have some major concerns about this work and its claimed take-home messages:

1. At no point, the authors measure the replicative lifespan of the constructed strains. I would recommend some single-cell microfluidic assays to measure the lifespan effects of key strains/conditions shown in this manuscript.

2. Almost all of the claims/conclusions are based on RNA-seq-measured small expression variations. It's OK to perform this bulk/high-throughput analysis first but it would be necessary to focus on key genes later one at a time at the protein level (promoter-GFP fusions perhaps). But which genes? That's one of the problematic areas of this manuscript, as the authors looked into the pairwise expression variations (the right thing to do) but did not see sufficiently high number of genes for downstream analysis. Perhaps not surprisingly, there is no specific gene(s) mentioned/claimed as the causal contributor to the shown effects. I was not very excited about their using small expression variations to reach the provocative conclusions of this work. 

3. Related to #2 above: Instead of small gene expression differences that are at the correlative/bulk level or SEP as the marker, there are some well-established yeast aging markers showing the pathology/physiology of yeast during its aging. It is absolutely necessary to look at least 2 such markers in the various strains the authors constructed. Equally importantly, the authors should perform live-cell tracking of aging yeast cells (carrying those markers one at a time) under the microscope and show how age affects their potential change in ERC-producing/absent/nonaccummulating cells. The same experiment should also be performed for the haploid strain which carries ChrXIIr on Chr5.

4. Studies aiming to track rDNA copy-number or its 3D dynamics are hard to perform due to genomic integration hardships (in a precise/countable manner) associated with its multi-copy nature. However, on the initially-nonrepetitive ChrXIIr, the authors can easily integrate a strong promoter driving a fluorescent reporter to the ChrXIIr region and see in live/aging cells how its expression (as a proxy for copy number changes) changes by replicative age. LacO arrays with LacI could also be integrated to observe the translocation dynamics of this region but perhaps that would be the aim of another paper.

---

## [Decision Letter · Decision Letter 2]

23 Jun 2023

Dear Jon,

Thank you for your patience while we considered your revised manuscript "Senescence in yeast is associated with chromosome XII fragments rather than ribosomal DNA circle accumulation" for publication as a Research Article at PLOS Biology. This revised version of your manuscript has been evaluated by the PLOS Biology editors, the Academic Editor and the original reviewers.

Based on the reviews, we are likely to accept this manuscript for publication, provided you satisfactorily address the remaining points raised by the reviewers and the following data and other policy-related requests.

IMPORTANT - Please attend to the following:

a) The team was discussing your Title; it's mostly clear, but I'm not sure that "chromosome XII fragments" quite captures it; also it was noticeable that in the Abstract you mention "amplification" but not "fragments." Looking at your Discussion, my understanding is that these are amplified fragments (large and linear) from that region of chr XII, yes? So I wonder if your title should read "...amplified linear fragments of chromosome XII..." and your Abstract should mention that they're linear fragments up to ~1.8 Mb in size?

b) Please attend to the remaining concerns raised by reviewers #2, #3 and #4. The Academic Editor confirms that minor revisions, mostly textual, should satisfy all reviewer concerns.

c) Please address my Data Policy requests below; specifically, we need you to supply the numerical values underlying Figs 1CD, 2ABCD, 3ABCD, 4ABCDEF, 5ABC, 6ABCDEFG, 7CD, S1AB, S2ABCD, S3ABCDEF, S4ABCDEFG, S5ABCDE, S6, S7ABCDEFG, S8BC, either as a supplementary data file or as a permanent DOI’d deposition. I note that you have already provided 5 supplementary data files, but the relationship between these files and the Figures is unclear; please provide all of the underlying data, clearly attributable to each Fig panel.

d) Please cite the location of the data clearly in all relevant main and supplementary Figure legends, e.g. “The data underlying this Figure can be found in S1 Data” or “The data underlying this Figure can be found in https://doi.org/10.5281/zenodo.XXXXX”

e) According to your expressed wishes, I've flagged to Production that these two papers should be scheduled for co-publication.

We expect to receive your revised manuscript within two weeks. 

*Published Peer Review History*

*Press*

Best wishes,

Roli

Roland Roberts, PhD

Senior Editor,

rroberts@plos.org,

PLOS Biology

DATA POLICY:

Regardless of the method selected, please ensure that you provide the individual numerical values that underlie the summary data displayed in the following figure panels as they are essential for readers to assess your analysis and to reproduce it: Figs 1CD, 2ABCD, 3ABCD, 4ABCDEF, 5ABC, 6ABCDEFG, 7CD, S1AB, S2ABCD, S3ABCDEF, S4ABCDEFG, S5ABCDE, S6, S7ABCDEFG, S8BC. NOTE: the numerical data provided should include all replicates AND the way in which the plotted mean and errors were derived (it should not present only the mean/average values).

We require the original, uncropped and minimally adjusted images supporting all blot and gel results reported in an article's figures or Supporting Information files. We will require these files before a manuscript can be accepted so please prepare and upload them now. Please carefully read our guidelines for how to prepare and upload this data: https://journals.plos.org/plosbiology/s/figures#loc-blot-and-gel-reporting-requirements

DATA NOT SHOWN?

REVIEWERS' COMMENTS:

Reviewer #1: 

[Accept; no comments]

Reviewer #2:

The authors have done an admirable job addressing reviewers concerns in this major revision of the manuscript. They took the concerns of the reviewers seriously and did their best job even if all concerns cannot be addressed. The message of the manuscript is quite interesting as it attempts to overturn a long held model in the field of yeast aging regarding causality of ERCs. I am still not convinced about the broad relevance to aging biology generally but that is only a minor detraction.

Because the revision is quite extensive I think it is natural to have a few new comments.

The authors have added a figure that contains a model for how the chromosome fragment impacts NPCs to create age related dysfunction. Several reviewers suggested this was needed. The model relies heavily on a previous publication. I'm not clear on why the fragment is predicted to attach to the NPC and create problems whereas normal chromosomes do not. Please clarify in the text.

The authors did not remove the original model figure regarding how the fragment forms in the first place. As stated previously, their data does not go very far to demonstrate the mechanism of formation so I was a bit surprised to see it still included although I imagine they have devoted substantial thought regarding the mechanism of formation. It makes for a somewhat unusual format with the conclusion being two model figures explaining two separate processes at the end of the manuscript. Although I do not think its essential to modify the manuscript, I would encourage the authors to consider whether they could either 1) move the mechanism of formation model to the supplement, or 2) combine the two models into a single integrated model that includes both formation and pathology.

Reviewer #3:

The authors have made a big effort to address my concerns and have added a lot of additional data to their study, especially regarding the role of slow growth and cell cycle arrest to SEP associated gene expression. I also appreciate that the authors note in the text that the gene expression data of rad52∆ mutants are outliers and in an effort to remedy these concerns they have added data for additional mutants to the manuscripts, even though some of these mutants suffer from the same issues as rad52 (involvement in DNA damage repair).

While I appreciate the authors effort, I still have some major concerns about the authors interpretation of the results. In particular, I don't see how their data supports two key conclusions:

1) The authors refute a role of ERCs in inducing cell senescence, based on the fact that mutants that do not form ERCs can still senesce. I agree with the authors interpretation that ERCs are not necessary for the SEP associated phenotypes (ERCs are not necessary). In fact, cells with irreparable DNA damage (rad52∆) might enter a terminal cell cycle arrest that will have a very similar molecular signature as cell senescence. The fact that cells can senesce without ERCs does not mean that ERCs do not contribute to SEP in wt cells. The plasmid accumulation experiment in Meinema et al 2022 suggest that indeed extrachromosomal plasmids are sufficient. The wording should be adjusted correctly.

2) The authors imply that amplification of XIIr causes SEP, but the data show a correlation, which does not prove causality. It is also possible that the partial aneuploidy is a consequence of impaired DNA damage repair and fatal cell cycle errors that are frequently observed long after cells have started to senesce (see Crane et al 2019 and Neurohr et al 2018). This interpretation fits well with the observation that rad52∆ mutants have increased levels of XIIr accumulation. The fact that XIIr does not accumulate to more than one copy per cell (which the authors suggest but this has not been analyzed) suggests that the translocation happens in the last division and not several divisions before cells stop dividing. In order to make the claim that XIIr contributes to the SEP transition (onset of slow cell divisions, which happens several divisions before cell death), the authors would at least need to show that the XIIr accumulation happens before or at the onset of the SEP, for example by using life span imaging with cells containing TetO repeats on the distal arm of chromosome XII. To establish causality, they would also need to show that preventing the translocation prevents SEP formation. As these would require quite substantial additional experiments, I suggest the authors tone down their language in regard to this model and present it as a possibility while pointing out that this is only one of several possible other interpretations.

Minor points:

3) By visual inspection it looks like the SEP associated correlation is mostly driven by the similarity between wild type and rad52∆ and the most important mutant spt3∆ (which is not short lived, has no ERCs but enters SEP) shows a much weaker correlation. Rather than comparing very different mutants and growth conditions it would be interesting to see the comparison of only wt and spt3 mutants, which have the most similar trajectories but differ in ERC levels.

4) The XIIr <-> V swap experiment is elegant, yet cells still senesce. This could be caused by accumulation of a large rDNA containing fragment as the authors argue, but it is not clear that this translocation still occurs as it is hard/impossible to distinguish from ERC accumulation. Is the control wt strain also an a/a diploid? The reason why this is important is that DNA damage repair pathways are regulated differently in haploid (a/a will behave as a haploid) and diploid (a/alpha) cells. (NHEJ is suppressed in diploids (a/alpha). 

Reviewer #4:

The authors have addressed most of my comments. Regarding my last point (#4), I disagree with them: they could have used the TEF1 promoter or the GAL1 promoter (constitutively expressed in gal80 deletion background) driving stable GFP, as strong promoters; they could have seen expression level differences on the order of 50% using these systems. Average reporter expression levels for similar-age cells across the population can be surprisingly stable across the healthy aging period (Sarnoski et al, 2018, cited already).

Overall, I believe the results from the manuscript will be of interest to the broad readership of PlosBiology and I warmly recommend its acceptance for publication.

---

## [Editor Report · Decision Letter 3]

12 Jul 2023

Dear Jon,

Thank you for the submission of your revised Research Article "Senescence in yeast is associated with amplified linear fragments of chromosome XII rather than ribosomal DNA circle accumulation" for publication in PLOS Biology. On behalf of my colleagues and the Academic Editor, Sarah Zanders, I'm pleased to say that we can in principle accept your manuscript for publication, provided you address any remaining formatting and reporting issues. These will be detailed in an email you should receive within 2-3 business days from our colleagues in the journal operations team; no action is required from you until then. Please note that we will not be able to formally accept your manuscript and schedule it for publication until you have completed any requested changes.

Sincerely,

Roli

Senior Editor

PLOS Biology

rroberts@plos.org